# EXPLOR: EXTRAPOLATORY PSEUDO-LABEL MATCHING FOR OOD UNCERTAINTY-BASED REJECTION

## ABSTRACT

EXPLOR is a novel framework that utilizes support-expanding, extrapolatory pseudo-labeling to improve prediction and uncertainty-based rejection on out-of-distribution (OOD) points. EXPLOR utilizes a diverse set of pseudo-labelers on an expansive augmented dataset to improve OOD performance through multiple MLP heads (one per pseudo-labeler) with shared embedding trained with a novel per-head matching loss. Unlike prior methods that rely on modality-specific augmentations or assume access to OOD data, EXPLOR introduces extrapolatory pseudo-labeling on latent-space augmentations, enabling robust OOD generalization with any real-valued vector data. In contrast to prior modality-agnostic methods with neural backbones, EXPLOR is model-agnostic, working effectively with methods from simple tree-based models to complex OOD generalization models. We demonstrate that EXPLOR achieves superior performance compared to state-of-the-art methods on diverse datasets in single-source domain generalization settings.

## 1 INTRODUCTION

It is well-known that the generalization capabilities of models can be severely limited when tested on out-of-distribution (OOD) data that deviates from the training-time distribution (Torralba and Efros, 2011; Liu et al., 2021; Freiesleben and Grote, 2023). This, in turn, affects many real-world applications where models may be evaluated on distribution-shifted data during deployment. For instance, these issues commonly arise in medical applications where patient distributions at inference time may deviate from the training data (Lee et al., 2023a). A potential strategy for the safe deployment of models in real-world applications is to employ novelty-based rejection (Dubuisson and Masson, 1993; Hendrickx et al., 2024), where predictions are rejected whenever the model is evaluated on an instance that deviates from the data distribution seen during training. While such approach is appropriate in certain scenarios (e.g., when a human can easily intervene upon rejection), this prevalent strategy is overly conservative as it foregoes any potential extrapolation[1] by design. That is, novelty-rejection forbids any form of extrapolation (predictions outside of the training data support), even when the model may be capable.

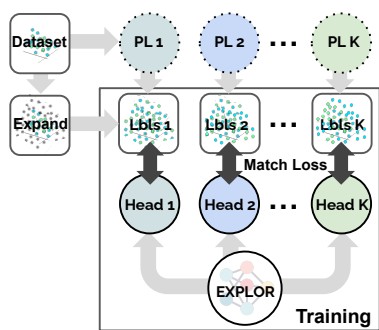

Figure 1: EXPLOR trains a multi-headed nnet with diverse pseudo-labelers (PLs) on expanded data.

**Motivation and Applications** Virtual screening in drug discovery (Shoichet, 2004) provides a driving application for this work. Here, models predict whether candidate molecules have desirable properties (e.g., binding to therapeutic protein targets) to filter large libraries of synthesizable compounds for empirical testing. Several challenges arise. First, the most valuable discoveries come from structurally novel molecules that differ substantially from those in the training set (Hu et al., 2017). Discovering the most innovative and novel drugs requires extrapolation beyond known scaffolds, yet traditional novelty-based rejection methods explicitly forbid such extrapolation (Dubuisson and Masson, 1993), ensuring reliability but fundamentally limiting discovery.

---

[1]We use the term extrapolation to encompass prediction outside of the training data distribution support.

Second, the utility of virtual screening diverges from the typical OOD generalization objective studied in machine learning (Yu et al., 2024). In practice, budgets only allow for a small fraction of candidates to be synthesized and tested, and the chemical space is so vast Hassen et al. (2025) that any training set covers only a tiny sliver. Thus, uniformly accurate predictions are neither realistic nor necessary. Instead, success depends on making high-confidence, high-precision predictions for the top-ranked candidates that will be selected for purchase. We quantify this via truncated precision/recall metrics eq. 1, reflecting the practical requirement of screening only the most promising molecules.

**Single-Source Setting** We target the single-source generalization setting (Qiao et al., 2020), which mirrors real-world workflows where only one labeled dataset—often from a narrow chemical space—is available for training. To address this, our **Ex**trapolatory **P**seudo-**L**abel Matching for **O**OD Uncertainty-Based **R**ejection (EXPLOR) framework trains a diverse set of pseudo-labelers on different feature/instance subsets, exposing the shared embedding to multiple views of the data. This encourages extrapolation while producing reliable confidence estimates, enabling robust and cost-effective drug discovery in unseen chemical domains.

**Goal** We focus on a model's ability to produce trustworthy high-confidence predictions on OOD points and reject unreliable predictions. To do so, we develop a *general*, modality- and model-agnostic end-to-end framework. To evaluate the model confidence, we propose a novel metric: area under precision/recall curve at recall less than $\tau$ (AUPRC@R$<\tau$). This metric specifically measures a model's ability to accurately predict positive examples in its most confident predictions (with relevance to chemoinformatic virtual screening). In contrast with prior work that depends heavily on modality-specific augmentations (e.g. for images (Yun et al., 2019), etc.) and/or the availability of multiple domains (Ding et al., 2022; Jang et al., 2023; Dou et al., 2019), our approach is fundamentally independent of data modality. Unlike prior modality-agnostic methods such as MODALS (Cheung and Yeung, 2021) that is modular or MAD (Qu et al., 2023) that is effective with complex neural architectures, EXPLOR is compatible with a broad spectrum of models and provides an end-to-end framework that not only generates extrapolated data but also effectively integrates them into training through diverse pseudo labeling via a novel per-head matching-based learning objective.

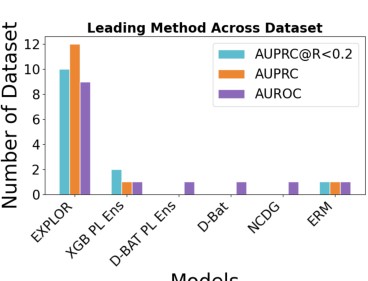

Figure 2: Tally of datasets where respective methods lead in metrics: AUPRC for recall < .2 (AUPRC@R< .2), AUPRC, & AUROC. (See further details in §4.)

**Contributions** In this work, we propose a new method for single source domain generalization – EXPLOR, designed to extrapolate effectively and yield reliable predictions in high-confidence regions through a novel training scheme using a multi-headed network that matches to diverse pseudolabels generated with expanded data (see Fig. 2).

Our key contributions are: (1) We develop a straightforward but effective strategy that yields a strong, diverse set of pseudo-labelers for self-training. (2) We propose a novel training loss for training multiheaded neural network architectures with pseudolabels—composed of a per-head matching loss and a mean-matching regularization loss—to ensure both diversity (via per-pseudo-labeler supervision) and consistency (via ensemble agreement). (3) We systematically evaluate models based on AUPRC@R$<\tau$, the normalized area under the precision recall curve below a conservative threshold $\tau$. While $R$ denotes recall in the metric name, we define the formal expression using a recall variable $r$:

$$\text{AUPRC@R}{<}\tau = \frac{1}{\tau} \int_0^\tau \text{Precision}(r)\, dr. \tag{1}$$

This novel metric measures a model's ability to predict the confidence of true positive examples in its most confident predictions (with relevance to virtual screening tasks). (4) We show state-of-the-art (SOTA) performance in prediction with a reject option based on estimated confidences, evaluated using AU{PR,RO}C-based metrics (see Fig. 1) in a single-source generalization setting (Qiao et al., 2020). (5) We conduct several ablations to better understand the keys to EXPLOR's success; moreover, by ablating the type of pseudolablers, we show EXPLOR's broad ability to improve over a model-agnostic set of experts.

## 2 RELATED WORK

**Domain Generalization** Domain generalization (DG) aims to learn a model that is able to generalize to multiple domains. A typical approach is to learn a domain invariant representation across multiple source domains. Domain invariant representation learning can be done by minimizing variations in feature distributions (Li et al., 2018; Ding et al., 2022) and imposing a regularizer to balance between predictive power and invariance (Arjovsky et al., 2019; Koyama and Yamaguchi, 2020). Another line of research incorporates data augmentation to improve generalizability. Basic transformations like rotation and translation, varying in magnitude, are commonly used on images to diversify the training data (Cubuk et al., 2019; Berthelot et al., 2020). More sophisticated augmentation techniques have recently surfaced: (Zhang et al., 2018) introduced mixup, which linearly combines two training samples; Yun et al. (2019) proposed CutMix, blending two images by replacing a cutout patch with a patch from another image; Zhong et al. (2022) adversarially augment images to prevent overfitting to source domains. We focus on augmentations that are general and applicable across modalities. Tian et al. (2023) introduced NCDG, which uses simple augmentations along with a loss function that maximizes neuron activity during training while minimizing standard classification loss. Their method minimizes the difference in the gradient of a coverage loss between standard training instances and augmented training instances. SAM (Foret et al., 2021) improves generalization by seeking parameters that lie in neighborhoods with uniformly low loss. UDIM (Shin et al., 2024) while finding flat loss parameters further enhances domain generalization by generating adversarial perturbations in latent space to expose and minimize inconsistencies between source domains and potential unseen target domains. While SAM and UDIM are modality-agnostic, our method EXPLOR is both model and modality-agnostic.

**Self-Training** Self-training uses an earlier model to pseudo-label unlabeled data, which is then added to the training set for subsequent model updates. Lee (2013) suggested a direct approach to retaining instances where the model has high prediction probabilities. Zou et al. (2018) proposed selecting a proportion of the most confident unlabeled points instead of using a fixed threshold. Later works combined pseudo-labeling with curriculum learning, adjusting class-wise thresholds over time to incorporate more informative samples (Cascante-Bonilla et al., 2020; Zhang et al., 2021).Another line of work improves pseudo-labeling robustness by promoting diversity in the labelers. Ghosh et al. (2021) used model ensembles as teachers, while Xie et al. (2019) added noise via Dropout (Srivastava et al., 2014) and data augmentation. FixMatch (Sohn et al., 2020) generates pseudo-labels from weakly augmented samples to supervise training on the corresponding strongly augmented samples. EXPLOR novelly leverages pseudo-labels by assigning each student head to a different expert, encouraging diversity, while aligns the ensemble prediction for consistency via a novel loss.

**Selective Classification** Reject option methods (or selective classification) aim to identify inputs where the model should abstain from predicting. Many approaches apply post hoc processing: after training, a rejection metric—such as the model's predicted probability—is computed, and predictions below a set threshold are rejected (Stefano et al., 2000; Fumera et al., 2000). Building upon these works, (Devries and Taylor, 2018) proposed to train a confidence branch alongside the prediction branch by incentivizing a neural network to produce a confidence measure during training; Geifman and El-Yaniv (2017) proposed a method for constructing a probability-calibrated selective classifier with guaranteed control over the true risk. Recently, methods adopting end-to-end training approaches have been proposed (Thulasidasan et al., 2019; Ziyin et al., 2019; Geifman and El-Yaniv, 2019). In these works, an extra class is added when predictions are made. If the extra class has the highest class probability for a sample, the sample is rejected. Most reject-option approaches are geared towards in-distribution rejection and utilize novelty-rejection when encountering any OOD points (Torralba and Efros, 2011; Liu et al., 2021; Freiesleben and Grote, 2023); instead, we propose to learn better conditional output probabilities on OOD data for more effective, capability-aware rejection.

**Ensemble Modeling** Ensembles utilize a diverse set of models jointly for better performance. Early methodologies for ensembles aggregate (bag) predictions from all models (Dietterich, 2007; Kussul et al., 2010) or a subset of the models in the ensemble (Jordan and Jacobs, 1993; Eigen et al., 2013). In the OOD setting, prior works addressed this problem by enforcing prediction diversity on OOD data (Pagliardini et al., 2023), ensembling moving average models (Arpit et al., 2022a), training an ensemble of domain specific classifiers (Yao et al., 2023), and training diverse model heads within a single network by maximizing disagreement on unlabeled OOD data (Lee et al., 2023b). EXPLOR adopts a multi-headed architecture that produces an ensemble to improve predictions on OOD data.

## 3 METHOD

Our approach, EXPLOR, consists of: (1) obtaining a diverse set of pseudo-labelers; (2) generating extrapolatory samples via latent space augmentation; and (3) training a multi-headed network to match diverse pseudo labels on both in-distribution (ID) and expanded data (one head per pseudo-labeler). Throughout, we assume the '*single-source*' generalization setting (Qiao et al., 2020), where we observe a single ID training dataset $\mathcal{D} = \{(x_i, y_i)\}_{i=1}^{N}$, and instances are drawn *iid* $(x_i, y_i) \sim \mathcal{P}_{\text{in}}$ *without* any accompanying environmental/domain/source information *nor any labeled/unlabeled OOD instances*. For simplicity, we write to the binary classification case, $y_i \in \{0, 1\}$, but our methodology is easily extendable to other supervised tasks. We design our method to work in general, non-modality specific[2] (e.g., image, text, audio) settings, i.e., $x_i \in \mathbb{R}^d$.

### 3.1 DIVERSE PSUEDO-LABELERS

EXPLOR leverages a set of diverse initial pseudo-labelers $\{g_k\}_{k=1}^{K}$, s.t. $g_k : \mathbb{R}^d \to \{0, 1\}$, to guide the training of a secondary model by providing pseudo-labels. There are many mixture of experts (Jordan and Jacobs, 1993; Eigen et al., 2013) and ensembling (Arpit et al., 2022a; Dietterich, 2007; Pagliardini et al., 2023; Yao et al., 2023) methods available , EXPLOR utilizes a collection of diverse pseudo-labelers by sub-selecting on both instance and feature subspace, specializing pseudo-labelers on distinct regions and views of the latent representation space.

### 3.2 EXPANSIVE AUGMENTATION OF TRAINING DATA

To train models capable of extrapolating to OOD samples, we need to expose them to data that lie outside the support of the training distribution. To reason about the support of the training data, and how to *expand* past it, we propose to leverage a latent factor space, $\varphi : \mathbb{R}^d \mapsto \mathbb{R}^s$. While learning semantically meaningful latent factor spaces remains an active area of research, we observed strong performance utilizing autoencoding techniques (see § 4), which carry a corresponding decoder $\gamma : \mathbb{R}^s \mapsto \mathbb{R}^d$. Without loss of generality, we consider centered latent spaces such that $\mathbb{E}[\varphi(X)] = 0$.

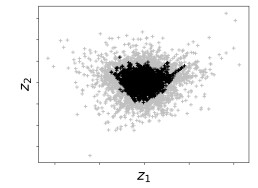

Figure 3: Expansion in latent space: points (black) are augmented (gray) and expand the distributional support.

We propose a novel, yet straightforward strategy to expand data outside of training distributional support: perturb instances to lie further away from the origin in latent space. In particular, if we have latent vector $z = \varphi(x)$, we propose to consider perturbations of the form $z' = (1 + |\epsilon|)z$ where $\epsilon \sim \mathcal{N}(0, \sigma^2)$, and one can utilize the decoder $x' = \gamma(z')$. I.e,, we define our expansion operation on a set of points as:

$$\mathbf{Ex}(\{x_i\}_{i=1}^{N}) \equiv \left\{ \gamma\left((1 + |\epsilon_i|)\varphi(x_i)\right) \mid \epsilon_i \sim \mathcal{N}(0, \sigma^2) \right\}_{i=1}^{N}. \qquad (2)$$

**Ex** will be a *stochastic* mapping. As shown in Fig. 3, our expansion covers areas away from the training support, even covering areas of OOD data. However, unlike with small jitter-based perturbations, where one can retain an original instance label, it is less clear how to derive an accompanying training signal for expansive augmentations. Below, we propose to leverage a pseudo-labeling scheme where we derive $K$ labels with the pseudo-labelers $(g_1(x'), \ldots, g_K(x'))$[3].

### 3.3 EXPLOR: EXTRAPOLATORY PSEUDO-LABEL MATCHING FOR OOD REJECTION

Once we generate extrapolated data via latent expansion, the key challenge becomes how to provide supervision on these OOD samples. Since true labels are unavailable, we leverage predictions from pseudo-labelers to pseudo label these points. To retain the diversity in the pseudo-labelers, we propose a multi-headed neural architecture trained via self-training. Specifically, the network consists of a shared multilayer perceptron (MLP), denoted as $\phi : \mathbb{R}^d \to \mathbb{R}^m$, which learns a common representation space, and $K$ labeler-specific heads, $h_1, \ldots, h_K$, where each head maps the shared representation to a prediction, e.g., $h_j : \mathbb{R}^m \to \mathbb{R}$ for logits in binary classification. Importantly,

---

[2]We avoid any modality or domain-specific augmentations.

[3]One may also train pseudo-labelers directly on the latent space, $(g_1(z'), \ldots, g_K(z'))$, and avoid the decoder.

each head is trained to match the output of a different pseudo-labeler, enabling the model to align with multiple diverse supervisory signals. The training loss is defined as a sum of per-head losses on a pseudo-labeled set $\mathcal{S}$, where each $h_j$ is optimized to match the corresponding pseudo-labeler's predictions. This encourages shared representation to support generalization across diverse pseudo-labeling sources. Our per head loss that matches pseudo labels provided by the experts to the MLP heads on a set $\mathcal{S}$ is as follow:

$$\mathcal{L}_{\text{match}}(\phi, \{h_j\}_{j=1}^K, \{g_j\}_{j=1}^K; \mathcal{S}) \equiv \frac{1}{|\mathcal{S}|K} \sum_{x \in S} \sum_{j=1}^K \ell(h_j(\phi(x)), g_j(x)), \tag{3}$$

where $\ell(\hat{y}, y)$ is a supervised loss (e.g., the cross-entropy loss). Moreover, we will utilize a mean-matching L1 loss

$$\mathcal{L}_{\text{mean}}(\phi, \{h_j\}_{j=1}^K, \{g_j\}_{j=1}^K; \mathcal{S}) \equiv \frac{1}{|\mathcal{S}|} \sum_{x \in S} \left| \frac{1}{K} \sum_{j=1}^K \sigma(h_j(\phi(x))) - \frac{1}{K} \sum_{j=1}^K g_j(x) \right|, \tag{4}$$

where $\sigma(\cdot)$ is the sigmoid. Our full EXPLOR loss is then:

$$\mathcal{L}_{\text{EXPLOR}}(\phi, \{h_j\}_{j=1}^K, \{g_j\}_{j=1}^K; \mathcal{D}) \equiv \mathcal{L}_{\text{mean}}(\phi, \{h_j\}_{j=1}^K, \{g_j\}_{j=1}^K; \mathcal{D}) \tag{5}$$
$$+ \mathcal{L}_{\text{match}}(\phi, \{h_j\}_{j=1}^K, \{g_j\}_{j=1}^K; \mathcal{D}) \tag{6}$$
$$+ \lambda \mathcal{L}_{\text{match}}(\phi, \{h_j\}_{j=1}^K, \{g_j\}_{j=1}^K; \mathbf{Ex}(\mathcal{D})). \tag{7}$$

Note that we provide additional supervisory losses on non-augmented $\mathcal{D}$ via $\mathcal{L}_{\text{mean}}$. Empirical results show (§ 4) that the network heads learn often learn comparably effective estimators to the pseudo-labelers, unlike with empirical risk minimization. However, we see more consistent improvements by not discarding the pseudo-labelers and bagging:

$$f_{\text{EXPLOR}}(x) \equiv \frac{1}{2K} \sum_{j=1}^K (g_j(x) + h_j(\phi(x))). \tag{8}$$

**Motivation** We expound on how EXPLOR may learn better estimates on OOD data through *diversity and multi-task learning*, and *variance reduction and regularization*.

*Diversity and Multi-task Learning.* In practice, we propose simple linear heads. At an intuitive level, this forces the MLP to learn a robust feature embedding that can 'mimic' the diverse views that the pseudo-labelers provide. That is, this will force the last hidden layer to featurize an embedding $\phi(x)$ that can, with simple linear projections, emulate a diverse set of labels. The per-head matching loss equation 3 may be formulated as a multi-task loss on a set a of $K$ *virtual* environments $\mathcal{E}_j(\mathcal{S}) = \{(x, g_j(x)) \mid x \in \mathcal{S}\}$: $\mathcal{L}_{\text{match}}(\phi, \{h_j\}_{j=1}^K, \{g_j\}_{j=1}^K; \mathcal{S}) = \frac{1}{K} \sum_{j=1}^K \mathcal{L}(h_j(\phi(\cdot)), \mathcal{E}_j(\mathcal{S}))$, where $\mathcal{L}(h_j(\phi(\cdot)), \mathcal{E}_j(\mathcal{S}))$ is the supervised loss on instances/labels in environment $\mathcal{E}_j(\mathcal{S})$ with estimator $h_j(\phi(\cdot))$ on the shared embedding $\phi$. Thus, when training on the expanded set of data-points, $\mathbf{Ex}(\mathcal{D})$, with pseudo-labels stemming from diverse labelers (e.g., trained on different subsets of features and instances), we see that our matching loss provides supervisory signals to learn: 1) on OOD data (through expansion); 2) robust embeddings that must generalize to diverse environments.

*Variance Reduction and Regularization.* Previous work has decomposed OOD generalization into bias/variance terms (Yang et al., 2020; Arpit et al., 2022b):

$$\mathbb{E}_{(x,y) \sim \mathcal{P}_{\text{out}}} \mathbb{E}_{\mathcal{D} \sim \mathcal{P}_{\text{in}}}[\text{CE}(y, f(x; \mathcal{D}))] = \mathbb{E}_{(x,y)}[\text{CE}(y, \bar{f}(x))] + \mathbb{E}_{x,\mathcal{D}}[\text{KL}(\bar{f}(x), f(x; \mathcal{D}))], \tag{9}$$

where CE is the cross-entropy loss, $f(x; \mathcal{T})$ is the model fit on dataset $\mathcal{T}$, $\bar{f}(x) = \mathbb{E}_{\mathcal{D}}[f(x; \mathcal{D})]$ is the expected prediction when averaging out draws on the (in-distribution) training dataset $\mathcal{D}$, and $\mathcal{P}_{\text{out}}$ is the OOD data distribution at inference time. Letting $\bar{g}(x) \equiv \frac{1}{K} \sum_{j=1}^K g_j(x)$, we may view $\bar{g}(x)$ as a bootstrap-like estimate for $\bar{f}(x)$. One may then take $\mathbb{E}_{x,\mathcal{D}}[\text{KL}(\bar{g}(x), f(x; \mathcal{D}))]$ as an approximation for $\mathbb{E}_{x,\mathcal{D}}[\text{KL}(\bar{f}(x), f(x; \mathcal{D}))]$ and roughly consider

$$\mathbb{E}_{(x,y) \sim \mathcal{P}_{\text{out}}} \mathbb{E}_{\mathcal{D} \sim \mathcal{P}_{\text{in}}}[\text{CE}(y, f(x; \mathcal{D}))] \approx \mathbb{E}_{(x,y)}[\text{CE}(y, \bar{f}(x))] + \mathbb{E}_{x,\mathcal{D}}[\text{KL}(\bar{g}(x), f(x; \mathcal{D}))], \tag{10}$$

which connects to equation 7 when interpreting our expanded points as a proxy for the (unknown) OOD distribution $\mathcal{P}_{\text{out}}$ and $\mathcal{L}_{\text{match}}(\phi, \{h_j\}_{j=1}^K, \{g_j\}_{j=1}^K; \mathbf{Ex}(\mathcal{E}))$ as a proxy for $\mathbb{E}_{x,\mathcal{D}}[\text{KL}(\bar{g}(x), f(x; \mathcal{D}))]$.

## 4 EXPERIMENTS

We conduct experiments on a varied set of real-world datasets to test the OOD generalizability of EXPLOR. We considered the single source domain generalization setting (e.g., (Qiao et al., 2020)), where our model is trained solely on ID data without any (labeled or unlabeled) OOD data during training/validation (e.g., precluding typical semi-supervised approaches), and without any accompanying environmental/domain/source information from ID training instances. Moreover, we note that we avoided utilizing any modality-specific information in EXPLOR (e.g., we do not utilize any domain specific augmentations) for generality. We performed two sets of experiments using different pseudo-labelers: one using 1024 XGB Classifiers (Chen and Guestrin, 2016) fit to random subsets of data instances and features as the diverse set of pseudo-labelers, and another using a complementary neural model consisting of 64 D-BAT (Pagliardini et al., 2023) networks trained in the same way to provide diverse neural pseudo-labels. For a fair/realistic evaluation, we avoided any hyper-parameter (e.g. number of training iterations, $\lambda$ in eq. 7) tuning on EXPLOR and utilized a fixed architecture of a 2-layer 512-ELU (Clevert et al., 2015) hidden-unit MLP with 1024 linear-output heads for the XGB pseudo-labelers and 64 heads for the D-BAT pseudo-labelers (please see other hyperparameters in Appx. B.1). For our latent space, we utilize PCA with 128 components as a linear analogue of an autoencoder. While OOD generalization is an active field of research (Freiesleben and Grote, 2023; Liu et al., 2021), methodology for general (non-modality specific) single source domain generalization is more limited. We provide context to our results through comparisons to a diverse set of existing strong domain generalization methods that approach the problem from various perspectives (and are applicable in the single-source setting).

**Single-Source Baselines**    In our experiments, we include three baselines that utilize data augmentation: AdvStyle (Zhong et al., 2022), Mixup (Zhang et al., 2018), and NCDG (Tian et al., 2023), two baselines that employ ensemble methods: D-BAT (Pagliardini et al., 2023) and EoA (Arpit et al., 2022a), as well as two methods that optimize robustness directly looking at the loss function: SAM (Foret et al., 2021) and UDIM (Shin et al., 2024). Mixup linearly combines two ID samples, AdvStyle adversarially augments ID data, NCDG takes a simple augmentation and optimizes neuron coverage, D-BAT enforces prediction diversity on OOD data, and EoA ensembles moving average models. SAM improves generalization by finding parameters that are in neighborhoods with uniformly low loss. UDIM generates adversarial perturbations in the latent space and seeks flat minima in the loss to improve generalization.

**Semi-Supervised Baselines**    We provide further context by comparing to semi-supervised methods for general tabular data, DivDis (Lee et al., 2023b) and FixMatch (Sohn et al., 2020). Note, these methods utilize more information than EXPLOR, and *have access to unlabeled OOD data from the test-time distribution*. That is, these approaches break from our single-source generalization setting and utilize additional information of the test-time distribution. DivDis is a two-stage framework that first trains diverse model heads using unlabeled target OOD data and then selects the best head with minimal supervision (i.e., using a single label from target OOD data). FixMatch generates filtered pseudo-labels using the model's predictions on weakly-augmented OOD unlabeled samples. The model is then trained to predict the pseudo-label when provided with a strongly augmented version of the same sample. As DivDis and FixMatch utilize additional information from single-source methods, we separate their results and denote them with an asterisk*.

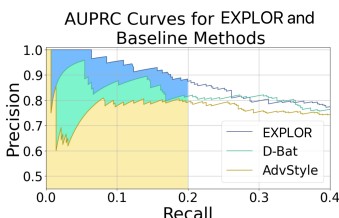

Figure 4: AUPRC at recall $< 0.2$ for EXPLOR and competitive baselines on hERG dataset.

**Metrics**    For prediction thresholding (rejection), we directly utilize the conditional probability $P(Y = 1 \mid X = x)$ generated by the models. Many real-world applications (e.g. drug virtual screening) utilize only high-confidence predictions. Thus, we paid close attention to high-confidence filtration and reported percent of AUPRC at conservative recall thresholds 'AURPC@R$<\tau$' (see eq. 1 for formal definition and Fig. 4 for illustration). We also reported AUPRC, AUROC. We report both pseudo-labeler ('PL ens') and EXPLOR (eq. 8) performance. Our code will be open-sourced upon publication.

### 4.1 CHEMICAL DATASETS

From ChEMBL(Gaulton et al., 2011) and Therapeutics Data Commons (TDC) (Huang et al., 2021), we collected the inhibition of human Ether-à-go-go-Related Gene (hERG), cytotoxicity of human A549 cells (A549_cells), agonists for Cytochrome P450 2D6 (cyp_2D6), and Ames mutagenicity (Ames) datasets. In these datasets, ID and OOD splits were determined based on the Murko scaffold of a molecule, a standard method to measure a model's ability to generalize to novel chemotypes, closely matching the real challenge of discovering active compounds. Moreover, we considered the "core ec50," "refined ec50," and "core ic50" ligand-based affinity prediction datasets (lbap) from DrugOOD (Ji et al., 2023) (the three hardest OOD performance gap datasets). For these datasets ID and OOD splits were determined based on the number of atoms in a molecule, such that larger molecules are considered the OOD set and smaller molecules, the ID set. Datasets are organized by domain and subsequently divided into training, OOD validation, and OOD testing sets in sequential order, testing generalization across size-driven chemical space shifts. For all datasets, we represented molecules using extended-connectivity fingerprints (Rogers and Hahn, 2010) with radius 2 (ECFP4) and with dimensionality 1024. ECFP4 is a standard method for molecular representation and was chosen for its simplicity in calculation as well as its ability to perform comparably to learned representations, such as those generated by graph neural networks on relevant classification tasks (Zagidullin et al., 2021).

**Results** We assessed models on ChEMBL, TDC, and DrugOOD datasets; our results are shown in Tab. 1. Across the chemical datasets, EXPLOR consistently outperforms both supervised and semi-

Table 1: Experiment results on ChEMBL (Gaulton et al., 2011), Therapeutics Data Commons (Huang et al., 2021), and DrugOOD (Ji et al., 2023) datasets. We **bold** best scores based on the mean minus 1 standard error. We *italicize* best scores when they are achieved by a semi-supervised method (*), that uses additional unlabeled OOD data during training.

| | | hERG | A549_cells | cyp_2D6 | Ames | refined ec50 val | refined ec50 test | core ic50 val | core ic50 test | core ec50 val | core ec50 test |
|---|---|---|---|---|---|---|---|---|---|---|---|
| AUPRC@R<0.2 | DivDis* | 85.65±2.37 | 89.45±1.16 | 79.98±1.10 | 95.80±0.34 | 96.02±0.19 | 86.51±0.28 | 97.80±0.07 | 89.72±0.06 | 93.86±0.42 | 80.18±0.92 |
| | FixMatch* | 79.66±0.57 | 90.81±4.30 | 83.58±1.98 | 93.55±0.64 | 94.48±0.47 | 86.38±0.35 | 97.62±0.06 | 92.89±1.36 | 95.02±0.76 | 70.49±0.43 |
| | ERM | 85.56±1.11 | 96.65±0.59 | 87.93±2.41 | 98.04±0.30 | 96.15±0.16 | 88.67±0.23 | 99.04±0.04 | 91.44±0.43 | 96.95±0.11 | 72.88±0.95 |
| | D-BAT | 84.48±1.74 | 98.26±0.14 | 91.40±0.98 | 99.04±0.24 | 96.97±0.16 | 88.78±0.40 | 98.13±0.08 | 91.79±0.38 | 93.81±0.22 | **84.35±1.35** |
| | AdvStyle | 88.21±0.77 | 97.77±0.28 | 84.83±0.93 | 99.05±0.17 | 95.13±0.13 | 88.21±0.37 | 97.04±0.17 | 89.05±0.22 | 94.84±0.31 | 84.51±2.36 |
| | EoA | 63.80±0.42 | 61.31±0.31 | 61.77±0.41 | 78.74±0.43 | 85.03±0.06 | 78.79±0.14 | 88.56±0.05 | 77.03±0.13 | 81.85±0.24 | 71.84±0.45 |
| | Mixup | 82.25±1.51 | 95.04±0.25 | 87.09±2.32 | 91.02±1.06 | 95.39±0.23 | 79.78±0.34 | 88.99±0.43 | 78.07±0.61 | 83.97±0.61 | 73.04±0.42 |
| | NCDG | 78.25±2.60 | 90.33±0.95 | 79.31±2.44 | 87.86±1.41 | 93.92±0.62 | 84.46±0.67 | 97.83±0.08 | 87.82±0.26 | 93.40±0.78 | 80.09±1.94 |
| | SAM | 84.37±1.06 | 95.70±0.75 | 85.02±0.69 | 96.66±0.38 | 96.79±0.17 | 87.50±0.23 | 98.78±0.09 | 89.40±0.40 | 95.02±0.37 | 71.92±1.71 |
| | UDIM | 85.07±2.32 | 95.87±0.36 | 84.80±0.84 | 96.53±0.61 | 96.97±0.13 | 88.16±0.78 | 98.61±0.12 | 90.13±0.38 | 94.05±0.30 | 71.37±1.00 |
| | D-BAT PL Ens | 84.86±0.82 | 97.45±0.32 | 92.56±0.81 | 99.47±0.14 | 96.67±0.06 | 88.11±0.04 | 98.74±0.07 | 94.05±0.08 | 97.51±0.21 | 69.93±0.04 |
| | XGB PL Ens | 94.44±0.17 | 98.22±0.08 | 95.51±0.19 | 97.84±0.20 | 98.00±0.07 | 89.48±0.24 | **99.14±0.02** | 94.20±0.15 | 97.79±0.11 | 68.48±0.30 |
| | **EXPLOR** (D-BAT) | 87.05±0.75 | 99.05±0.12 | 95.33±0.51 | **99.87±0.16** | 97.78±0.04 | 88.97±0.02 | 99.10±0.16 | **94.52±0.02** | 95.25±0.01 | 75.50±0.15 |
| | **EXPLOR** (XGB) | **94.67±0.29** | **98.87±0.09** | **96.88±0.25** | 98.45±0.19 | **98.45±0.06** | **89.76±0.26** | 99.15±0.05 | 94.42±0.09 | **98.66±0.10** | 69.04±0.50 |
| AUPRC | DivDis* | 67.70±0.25 | 76.45±0.63 | 65.76±0.45 | 82.37±0.27 | 89.62±0.12 | 80.92±0.06 | 93.34±0.10 | 81.48±0.30 | 83.85±0.41 | *75.09±0.38* |
| | FixMatch* | 45.16±3.11 | 51.31±1.28 | 32.65±2.16 | 72.54±1.15 | 87.14±0.58 | 81.51±0.31 | 92.38±0.15 | 82.78±0.57 | 80.92±1.12 | 70.57±0.32 |
| | ERM | 68.77±0.40 | 81.73±0.37 | 67.97±0.81 | 86.01±0.23 | 89.84±0.07 | 82.26±0.07 | 94.72±0.07 | 82.59±0.21 | 87.70±0.09 | 70.95±0.44 |
| | D-BAT | 54.60±1.59 | 67.04±0.53 | 47.42±0.85 | 70.44±0.58 | 84.70±0.51 | 70.08±0.69 | 90.84±0.28 | 73.45±0.90 | 76.64±0.49 | 54.87±0.99 |
| | AdvStyle | 51.54±0.93 | 65.02±0.72 | 44.41±0.96 | 74.98±0.49 | 83.01±0.90 | 69.48±2.16 | 88.54±1.44 | 72.11±1.49 | 81.17±1.72 | 58.40±2.18 |
| | EoA | 43.30±0.51 | 44.95±0.24 | 37.37±0.95 | 59.43±0.18 | 69.66±0.47 | 57.71±0.79 | 79.12±0.09 | 56.52±0.42 | 64.16±0.51 | 36.50±1.48 |
| | Mixup | 42.42±0.85 | 50.52±0.50 | 27.79±1.49 | 60.94±0.80 | 80.36±0.88 | 72.88±1.67 | 86.88±0.14 | 74.99±0.15 | 73.03±1.67 | 60.84±4.31 |
| | NCDG | 65.03±0.75 | 79.50±0.79 | 65.31±1.00 | 76.48±0.30 | 89.27±0.22 | 80.54±0.21 | 94.17±0.08 | 81.19±0.11 | 87.72±0.37 | 74.99±0.74 |
| | SAM | 66.89±0.28 | 79.88±0.34 | 67.37±0.31 | 82.50±0.39 | 89.86±0.09 | 81.93±0.20 | 94.16±0.10 | 81.03±0.23 | 86.66±0.20 | 70.92±0.40 |
| | UDIM | 67.23±0.60 | 79.98±0.37 | 66.86±0.43 | 82.97±1.05 | 89.95±0.11 | 82.19±0.41 | 94.05±0.10 | 81.29±0.25 | 86.35±0.15 | 71.20±0.35 |
| | DBAT PL Ens | 72.22±0.18 | 83.79±0.31 | 73.26±0.65 | 89.08±0.11 | 91.19±0.08 | 82.72±0.04 | 94.79±0.10 | 84.11±0.01 | 88.36±0.28 | 72.13±0.07 |
| | XGB PL Ens | 72.19±0.07 | 84.10±0.01 | 72.93±0.13 | 87.43±0.02 | 91.21±0.02 | 82.61±0.05 | 94.91±0.03 | 84.13±0.06 | 88.44±0.05 | 71.80±0.07 |
| | **EXPLOR** (D-BAT) | 72.73±0.33 | **84.80±0.11** | 73.64±0.45 | **89.38±0.06** | 90.56±0.03 | 82.95±0.06 | 94.89±0.02 | 84.11±0.01 | 88.44±0.02 | **73.13±0.02** |
| | **EXPLOR** (XGB) | **73.26±0.08** | 84.60±0.08 | **73.59±0.15** | 88.50±0.10 | **91.59±0.03** | **83.06±0.10** | **95.38±0.01** | **84.77±0.02** | **89.52±0.05** | 71.41±0.11 |
| AUROC | DivDis* | 71.19±0.57 | 72.20±0.41 | 64.50±0.77 | 77.05±0.47 | 65.41±0.81 | 58.53±0.46 | 66.68±0.28 | 57.04±0.08 | 73.23±0.40 | 61.15±0.50 |
| | FixMatch* | 68.41±0.84 | 61.16±1.31 | *80.85±0.40* | 70.32±0.59 | 70.32±0.59 | 52.69±0.69 | 66.37±0.50 | 58.37±0.58 | 74.38±0.16 | 62.05±0.55 |
| | ERM | 73.58±0.23 | 76.58±0.30 | 65.26±0.55 | 82.02±0.26 | 67.73±0.18 | 59.15±0.23 | 77.41±0.24 | 62.68±0.31 | 72.24±0.09 | 52.32±0.59 |
| | D-BAT | 76.58±0.45 | 78.16±0.23 | 67.54±0.47 | 83.82±0.15 | 75.26±0.28 | 58.21±0.26 | 72.09±0.19 | 60.32±0.25 | **80.31±0.08** | 64.82±0.18 |
| | AdvStyle | 75.84±0.46 | 76.13±0.28 | 65.51±0.62 | 85.56±0.71 | 75.97±0.39 | 58.86±0.25 | 70.78±0.35 | 59.62±0.30 | 78.36±0.23 | 64.14±0.27 |
| | EoA | 68.02±0.34 | 68.33±0.24 | 60.50±0.48 | 74.77±0.25 | 64.91±0.34 | 52.71±0.44 | 59.27±0.20 | 54.63±0.24 | 62.99±0.16 | 55.83±0.18 |
| | Mixup | 73.96±0.25 | 76.57±0.42 | 67.53±0.90 | 78.43±0.49 | 68.20±0.66 | 56.33±0.45 | 60.39±0.40 | 56.50±0.37 | 64.24±1.23 | 57.75±0.80 |
| | NCDG | 70.76±0.20 | 76.45±0.65 | 63.60±0.62 | 73.39±0.68 | 73.66±0.53 | 57.70±0.81 | 67.20±0.33 | 57.18±0.18 | 76.48±0.22 | 61.44±0.13 |
| | SAM | 72.17±0.41 | 74.67±0.49 | 64.45±0.53 | 78.51±0.47 | 67.33±0.22 | 58.97±0.40 | 75.63±0.35 | 60.07±0.35 | 71.06±0.27 | 52.50±0.36 |
| | UDIM | 73.17±0.38 | 74.64±0.38 | 64.09±0.66 | 78.63±1.30 | 67.50±0.29 | 59.30±0.63 | 75.38±0.34 | 60.29±0.39 | 71.37±0.18 | 53.43±0.43 |
| | DBAT PL Ens | 76.51±0.07 | 78.34±0.35 | 70.80±0.36 | 84.37±0.08 | 69.40±0.06 | 60.04±0.31 | 77.83±0.40 | 64.66±0.04 | 74.51±0.48 | 56.39±0.20 |
| | XGB PL Ens | 74.74±0.06 | 79.17±0.02 | 70.33±0.07 | 81.87±0.04 | 73.70±0.07 | 56.48±0.06 | 70.17±0.04 | 59.77±0.02 | 77.78±0.09 | 64.89±0.06 |
| | **EXPLOR** (D-BAT) | **76.84±0.10** | 79.27±0.14 | 70.11±0.33 | **85.18±0.04** | 70.64±0.04 | **59.90±0.03** | **78.52±0.37** | **64.67±0.01** | 75.34±0.07 | 56.32±0.03 |
| | **EXPLOR** (XGB) | 75.97±0.08 | **79.53±0.06** | 70.54±0.42 | 83.78±0.12 | **75.47±0.09** | 55.40±0.20 | 71.28±0.10 | 60.63±0.17 | 80.01±0.03 | **66.14±0.05** |

supervised baselines and consistently shows gains over its pseudo-labeler methods (see results using other pseudo-labelers in Appx.C.5). Note further, that EXPLOR's performance gain is especially significant at conservative recalls (AUPRC@R<0.2, e.g., see Fig. 4), which indicates that EXPLOR outperforms other models in virtual screening tasks to filter drug candidates (where false positives would lead to wasted resources). Please see below (§ 5), for further analysis on the improved performance of EXPLOR w.r.t. increased confidence (Fig. 5) and pseudo-labeler diversity (Fig. 7). We examine the mean variance (reflecting greater heterogeneity) of fingerprint features for selected OOD instances with predicted confidence>0.9 on the 3 ChEMBL sets: EXPLOR (**0.39**), D-BAT (0.33), EoA (0.30), AdvStyle (0.34), Mixup (0.37), and NCDG (0.23). Thus, EXPLOR is confidently selecting structurally diverse compounds rather than to a narrow subset of the chemical space.

## 4.2 OTHER REAL WORLD DATASETS

Next, we further evaluate our method in non-chemical domains across a diverse range of real-world OOD scenarios using the Tableshift datasets (Gardner et al., 2023). We selected a diverse collection of Tableshift datasets, based on unrestricted availability and in/out-of-domain performance discrepancy, coverings areas including: finance, education, and healthcare. Each dataset has an associated real-world shift and a related prediction target (see Gardner et al. (2023) for further details). Results on the Tableshift are shown in Tab. 7. As before, we consider the same single-source domain generalization setting. We can see that even over diverse applications, our EXPLOR method is able to perform well and often outperforms competitive baselines. Moreover, eventhough DivDis and FixMatch use additional unlabeled OOD data, EXPLOR outperforms both on the majority of datasets, highlighting EXPLOR's ability for generalization on completely *unforeseen* OOD instances (without any knowledge of test time distribution).

Table 2: Experiment results on Tableshift (Gardner et al., 2023) datasets. We **bold** best scores based on the mean minus 1 standard error.

| | | Childhood Lead | FICO HELOC | Hospital Readmission | Sepsis |
|---|---|---|---|---|---|
| AUPRC@ R<0.2 | DivDis[*] | 89.77±2.51 | 85.50±2.65 | 83.22±2.57 | 60.66±1.57 |
| | FixMatch[*] | 81.75±3.69 | 77.76±1.94 | 67.84±1.95 | 17.28±0.58 |
| | ERM | 43.66±0.00 | 85.74±0.78 | **84.35±0.28** | 15.30±0.53 |
| | D-BAT | 62.82±0.00 | 91.20±0.24 | 78.84±0.12 | 75.37±0.38 |
| | AdvStyle | 64.96±0.01 | 88.71±0.65 | 72.91±0.58 | 59.83±0.63 |
| | EoA | 77.43±0.97 | 59.53±2.24 | 51.83±1.66 | 41.10±1.56 |
| | Mixup | 50.00±0.00 | 91.16±2.10 | 69.19±3.95 | 66.09±1.15 |
| | NCDG | 42.49±1.31 | 87.93±2.35 | 68.88±1.15 | 15.22±0.38 |
| | SAM | 95.00±0.00 | 91.42±0.69 | 74.13±0.49 | 65.34±0.73 |
| | UDIM | 94.11±0.90 | 92.58±0.67 | 73.65±0.67 | 65.09±0.28 |
| | DBAT PL Ens | 93.81±0.03 | 92.87±0.05 | 73.52±0.37 | 72.74±0.13 |
| | XGB PL Ens | 97.39±0.06 | 90.07±0.32 | 58.30±0.09 | **78.62±1.57** |
| | **EXPLOR** (D-BAT) | 94.69±0.02 | **93.33±0.02** | 77.20±0.23 | 73.89±0.08 |
| | **EXPLOR** (XGB) | **97.92±0.20** | 91.72±0.62 | 67.57±0.12 | 76.95±1.45 |
| AUPRC | DivDis[*] | 76.33±0.86 | 82.47±1.05 | 65.95±2.15 | 58.85±1.21 |
| | FixMatch[*] | 75.39±0.58 | 72.22±4.46 | 56.54±1.21 | 18.24±0.66 |
| | ERM | 23.60±0.11 | 77.38±0.27 | **67.56±0.13** | 11.12±0.12 |
| | D-BAT | 71.85±0.01 | 80.91±0.17 | 63.29±0.08 | 58.17±0.21 |
| | AdvStyle | 48.37±2.77 | 79.63±1.65 | 38.13±3.80 | 54.34±0.27 |
| | EoA | 49.48±0.19 | 59.53±2.24 | 29.45±4.95 | 11.21±2.21 |
| | Mixup | 50.00±0.00 | 80.95±0.63 | 14.15±1.06 | 56.80±0.40 |
| | NCDG | 23.08±0.24 | 79.05±1.21 | 58.95±0.23 | 11.96±0.32 |
| | SAM | 75.00±0.00 | 81.61±0.43 | 61.12±0.28 | 57.90±0.22 |
| | UDIM | 73.70±1.38 | 82.10±0.44 | 60.95±0.33 | 57.97±0.17 |
| | DBAT PL Ens | 82.72±0.01 | 81.35±0.03 | 62.34±0.14 | 59.97±0.10 |
| | XGB PL Ens | 86.39±0.19 | 83.80±0.07 | 62.83±0.03 | **64.01±0.94** |
| | **EXPLOR** (D-BAT) | 82.76±0.02 | 81.37±0.04 | 63.30±0.14 | 50.97±0.05 |
| | **EXPLOR** (XGB) | **86.70±0.28** | **84.02±0.09** | 63.62±0.08 | 62.21±0.52 |
| AUROC | DivDis[*] | 77.74±1.90 | 83.55±1.06 | 65.28±0.59 | 62.33±0.61 |
| | FixMatch[*] | 79.87±0.29 | 74.29±1.69 | 55.78±0.98 | 49.50±1.01 |
| | ERM | 78.41±0.14 | 73.71±0.17 | **67.06±0.10** | 62.79±0.14 |
| | D-BAT | 79.13±0.02 | 76.13±0.03 | 63.22±0.03 | 57.95±0.04 |
| | AdvStyle | 74.45±0.04 | 77.23±1.69 | 61.32±0.34 | 55.31±0.34 |
| | EoA | 72.62±0.32 | 54.67±2.55 | 51.65±1.70 | 49.14±2.60 |
| | Mixup | 50.00±0.00 | 78.74±0.17 | 63.37±0.25 | 56.82±0.26 |
| | NCDG | 76.91±0.13 | 75.09±0.79 | 58.67±0.16 | **63.41±0.93** |
| | SAM | 50.00±0.00 | 79.70±1.64 | 61.14±0.32 | 58.71±0.23 |
| | UDIM | 54.50±4.18 | 79.41±1.07 | 61.05±0.32 | 58.87±0.24 |
| | DBAT PL Ens | 84.89±0.05 | 76.32±0.03 | 63.16±0.10 | 59.23±0.04 |
| | XGB PL Ens | 84.88±0.19 | **83.53±0.03** | 63.18±0.03 | 62.53±0.82 |
| | **EXPLOR** (D-BAT) | 84.36±0.23 | 76.27±0.01 | 63.51±0.16 | 59.01±0.07 |
| | **EXPLOR** (XGB) | **87.95±0.25** | 83.11±0.04 | 63.65±0.07 | 61.42±0.35 |

## 4.3 ABLATION STUDIES

**Per-head Matching Ablation** Next, we perform ablation on the per-head matching loss scheme (eq. 7) used in EXPLOR with a simpler alternative: mean-only matching (MM) (eq. 5) on the expanded points. This ablation tests whether the diversity induced by per-head supervision aids performance. Here we utilized single headed (SH) and multi-headed (MH) MLPs with mean matching, where EXPLOR uses multi-headed MLP with per head loss that matches network heads to diverse pseudo-labels (see Appx. C.6 for details on the alternative losses). EXPLOR achieves the highest AUPRC improvements ($\Delta$) over pseudo-labelers (+1.11 at R<.2, +1.03 at R<1) compared to SH+MM (+0.59, +0.45) and MH+MM (+0.25, +0.55), showing evidence for the effectiveness of training our embedding through the diverse multi-task loss induced by per-head matching (§ 3).

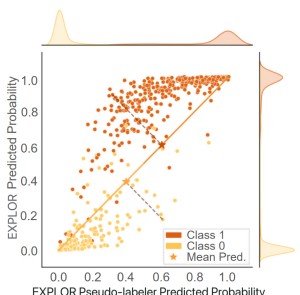

**Bottleneck Ablation** We motivated EXPLOR's performance in terms of a multi-task scheme, which encourages the bottlenecks (the last hidden layer) to learn an embedding that can generalize to mimic predictions on a diverse set of pseudo-labels (§ 3). We test this motivation by comparing the original architecture (Full: $2 \times 512$ hidden layer) with a stronger bottleneck still, a small architecture (Tiny: $2 \times 32$ hidden layer). (See Appx. C.7 for more details and results.) The performance gap is marginal between the 'Full' and 'Tiny' model (a $0.86\%$ difference) when using our proposed loss. In contrast, when using empirical risk minimization, we observe a $4.95\%$ performance drop. This suggests that the bottlenecking properties of our method are key to EXPLOR's performance, and show promise for EXPLOR in resource-constrained settings.

Figure 5: *Predicted probabilities from EXPLOR network and pseudo-labeler.* We highlight example instances where the base experts initially makes incorrect predictions but are corrected when we average the predicted probabilities from EXPLOR network and EXPLOR pseudo-labeler.

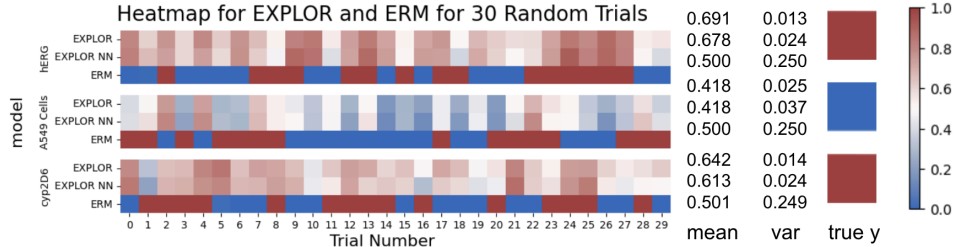

Figure 6: Example predicted probabilities an OOD data instance for ChEMBL datasets. Predictions are shown for a most variant instance, under empirical risk minimization (ERM), for each dataset over the 30 trials. EXPLOR (eq. 8), EXPLOR multi-headed neural network, and ERM were trained were on 30 subsamples of ID Data. We can see that not only do EXPLOR models reduce the variance for these challenging instances, but they are able to produce the correct label prediction (unlike ERM).

**Variance Reduction Across Bootstrapped Models** To empirically validate the variance reduction effect discussed in § 3.3, we trained 30 bootstrap-type trials using an ERM and EXPLOR model loss (each using a random subsamples of size 1,000 of the ID training data) and evaluated the variance of predicted probabilities on the same OOD held-out points.

Table 3: Predictive variance across 30 bootstrap-trained models on OOD Data. Lower values indicate more stable predictions under distribution shift.

| Method | hERG | A549 Cells | cyp2D6 |
|---|---|---|---|
| ERM | 0.117 | 0.119 | 0.156 |
| EXPLOR NN | 0.023 | 0.028 | 0.025 |
| EXPLOR | **0.015** | **0.019** | **0.016** |

Tab. 3 reports the mean predictive variance across OOD samples for ChEMBL datasets. The ERM trained neural network exhibits high instability over predictions for the OOD data, whereas EXPLOR-NN, our multi-headed architecture trained using eq. 7 yields a substantial reduction in variance (Fig.6). Full EXPLOR model (also ensembling with the original pseudo-labler) further decreases variance across all datasets, achieving 6–8 times lower variance

than ERM. These results provide direct empirical evidence that EXPLOR produces markedly more stable and reliable predictions on unseen chemical scaffolds, supporting the variance-reduction motivation introduced in § 3.3.

## 5 CONCLUSION

EXPLOR addresses the challenging single source OOD generalization problem for real-valued vector data, outperforming baselines, including those that leverage unlabeled OOD data. Across all metrics/datasets, EXPLOR achieves a leading tally of 31, far exceeding the pseudo-labeler methods and other baselines as shown in Fig. 1. Despite its performance, EXPLOR does not incur an out-sized computational cost; the training of pseudo-labelers may be done in parallel, and the cost of expanding data in the latent space is negligible (see Appx. B.3 for additional timing details).

In summary, EXPLOR presents a simple yet powerful model and data-modality agnostic approach to OOD generalization under single-source setting via pseudo-labels and head-specific matching. Each pseudo-labeler is paired with a dedicated prediction head in the neural network, and the training objective explicitly aligns each head with the pseudo-labels it receives. EXPLOR's architecture yields robust embeddings and demonstrates superior performance in regions characterized by high-confidence OOD predictions. We believe this work opens promising directions for general-purpose, modality- and model- agnostic OOD learning, particularly in high-stakes applications like drug discovery, where confident extrapolation is critical and labeled data is scarce.

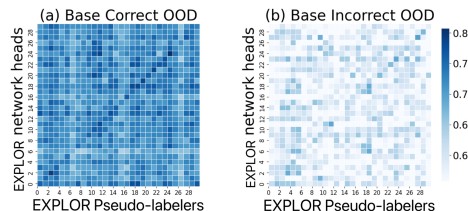

Figure 7: Experts correlations between EXPLOR pseudo-labelers and EXPLOR network heads on "Ames" dataset. (a) EXPLOR experts have a high correlation with EXPLOR pseudo-labelers on sample the pseudo-labelers make correct predictions. (b) EXPLOR experts show low correlations with EXPLOR pseudo-labelers on samples where the pseudo-labelers make incorrect predictions.

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

APPENDIX

# A    USAGE OF LLMS STATEMENT

In this submission, LLMs were used only as a general editing tool. Part of the text were drafted by authors and fed into LLMs for grammar check and help polish the text.

# B    ADDTIONAL EXPERIMENT DETAILS

## B.1    EXPLOR TRAINING DETAILS

In all of our experiments we used the Adam (Kingma and Ba, 2014) optimizer and mini-batches of size 256. One Nvidia A100 GPU with 40GB GPU memory was used to run our experiments, and duration for model training is approximately 0.5 hours. $\lambda = 0.5$ was used for the $\mathcal{L}_{\text{match}}$ for the expanded points. As noted in Sec. 3.2 we trained the EXPLOR models directly in the latent space to avoid the need for the decoder (and also allowed baselines to do this if it aided their performance). In the experiments on hERG, A549_cells, CYP_2D6, Ames, core ec50, refined ec50, EXPLOR was trained for 10000 iterations. Arithmetic mean between EXPLOR and pseudo-labeler ensemble was reported. We performed 5 trails on each of the datasets for EXPLOR.

## B.2    BASELINE SETUP

We implemented all baselines we are comparing against EXPLOR following the implementation details in their paper and/or using Github implementations (if available). Since the fingerprints representation of chemicals are quite sparse, we preformed dimension reduction using PCA with 128 components on all chemical datasets. For D-BAT (Pagliardini et al., 2023) with existing implementations designed for tabular data, we utilized their original model architectures. For the other baseline methods without implementation specifically for tabular data, we adopted a structure comprising two 512 ELU(Clevert et al., 2015) layers to closely mimic the EXPLOR network architecture. The Adam (Kingma and Ba, 2014) optimizer was used for training baseline models.

**ERM** We implement a multiheaded ERM baseline that follows the EXPLOR neural network architecture. The architecture uses the same shared feature extractor followed by a 1024-dimensional output layer, where each output corresponds to an independent binary classifier. We train for 10,000 iterations with a learning rate of 0.0005 and maintain a moving average model updated every 2,500 iterations.

**D-BAT** In our experiments, the D-Bat(Pagliardini et al., 2023) models used MLP architecture with one 128 LeakyRelu(Maas, 2013) layer following the architecture in their Github. Their paper (Pagliardini et al., 2023) discussed two settings, and we focused on the scenario where perturbation data differs from the distribution of test data, adhering to the single-source domain generalization setting. We trained an ensemble of five models sequentially for the D-bat baseline models and the predictions from the 5 models were averaged to obtain the final prediction.

**EoA** We trained an ensemble of 5 simple moving average model following the method described in (Arpit et al., 2022a). We start calculating the moving average at iteration 50 and trained the models for 200 iterations. The predictions from the 5 models were averaged to obtain the final prediction for EoA.

For **AdvStyle** (Zhong et al., 2022) and **Mixup** (Zhang et al., 2018), the methodologies were straightforward. We experimented with training using various numbers of iterations and reported the most promising results. Note that we used alpha=0.7 when combining the 2 samples for Mixup. We executed all baseline experiments five times on each dataset to ensure a precise estimation of performance.

**DivDis** We utilized all unlabeled target OOD data for training and a single label from this data for supervision. An ensemble of 5 models was trained for 100 iterations with early stopping, and each model has 2 classification heads. Across all datasets, we set $\lambda_1 = 10$ (encouraging disagreement among model heads), while $\lambda_2$ (an optional hyperparameter prevents degenerate solutions) was set to

Table 4: Training Time for EXPLOR and baseline methods.

|          | wall clock (m) |
|----------|----------------|
| EXPLOR   | 4.2            |
| SAM      | 1.13           |
| UDIM     | 2.07           |
| Dbat     | 3.5            |
| EoA      | 67.9           |
| Advstyle | 4.8            |
| Mixup    | 1.2            |
| NCDG     | 2.7            |
| ERM      | 0.9            |

0 for DrugOOD and ChEMBL and to 10 for TableShift. The final prediction is the average of the 5 models' predictions.

**FixMatch** Originally, Fixmatch (Sohn et al., 2020) was designed for image data, so the sense of weak and strong augmentations were image based. To adapt the method to our modality agnostic setting we used $x * (1 + \alpha)$ as the weak augmentation, $x * (1 + 2 * \alpha)$ as strong augmentation, where $\alpha$ is a small noise drawn from the standard normal distribution.

For **NCDG** Tian et al. (2023), we adapt the method to use the EXPLOR architecture (rather than a ResNet model) and the EXPLOR augmentation method. We set t=0.005 (the threshold for neuron activation in coverage computation), $\lambda = 0.1$ (the weight coefficient for neuron coverage loss), and $\beta = 0.01$ (the weight for gradient similarity regularization loss). Five trials were run on each dataset and averaged to obtain the final results.

**SAM** We train the MLP with 2 hidden layers of size 512 (same hidden layer as EXPLOR) using the SAM (Foret et al., 2021) objective. We used a $\rho = 0.05$ (radius for evaluating the loss sharpness) and $\epsilon - 1e^{-5}$ (perturbation weight). The model was trained for 100 epochs with a learning rate of 0.001.

**UDIM** We train the MLP with 2 hidden layers of size 512 (same hidden layer as EXPLOR) using the UDIM (Shin et al., 2024) framework. We used a $\rho = 0.05$ (radius for evaluating the loss sharpness), $\rho_x = 0.5$ (radius for adversarial perturbations), and $\lambda = 0.5$ (domain inconsistency regularizer weight). The model was trained for 100 epochs with a learning rate of 0.001.

### B.3 EXPLOR TRAINING TIME

In Tab. 4, we report the training time for EXPLOR and baseline models we are considering. Note that the pseudo-labelers can be training in parallel with enough computational resources (and are each quick to train at $< 1s$).

## C ADDITIONAL EXPERIMENT AND ABLATION RESULTS

### C.1 FULL EXPERIMENT RESULTS ON CHEMBL AND THERAPEUTICS DATA COMMONS

In Tab. 6, we report the full results on hERG, A549_cells, cyp_2D6. and Ames.

### C.2 FULL EXPERIMENT RESULTS ON DRUGOOD

In Tab. 5, we report the full results on core ec50, refined ec 50, and core ic50 from DrugOOD (Ji et al., 2023).

### C.3 DIVERSITY OF PREDICTIONS

In drug discovery applications, models should predict on structurally diverse compounds. To assess the diversity of model behavior in high confidence out-of-distribution (OOD) predictions, we examine the average variance of fingerprint features for instances with predicted confidence greater than 0.9 on the 3 ChEMBL datasets (var@p>0.9). Higher variance reflects greater heterogeneity among the selected molecules. We observe the following var@p>0.9 on ChEMBL datasets: EXPLOR (**0.391**), D-BAT (0.337), EoA (0.301), AdvStyle (0.349), Mixup (0.370), and NCDG (0.239). That is,

Table 5: Full experiment results on DrugOOD datasets. We **bold** best scores based on the mean minus 1 standard deviation. Note that * refers to a semi-supervised method.

| | | refined ec50 val | refined ec50 test | core ic50 test | core ic 50 test | core ec50 val | core ec50 test |
|---|---|---|---|---|---|---|---|
| AUPRC@ R<0.1 | DivDis* | 96.63±0.54 | 88.87±0.58 | 98.41±0.26 | 91.72±1.47 | 98.48±0.06 | 75.02±1.11 |
| | FixMatch* | 94.78±0.63 | 86.53±0.25 | 97.70±0.05 | 95.89±1.46 | 96.42±1.06 | 69.80±0.25 |
| | ERM | 97.76±0.14 | **90.40±0.27** | 99.38±0.05 | 93.81±0.55 | 96.21±1.02 | 82.45±2.93 |
| | D-BAT | 97.19±0.19 | 88.93±0.48 | 98.25±0.09 | 91.89±0.39 | 94.04±0.25 | **86.59±1.42** |
| | AdvStyle | 96.37±0.32 | 88.69±0.43 | 98.10±0.17 | 89.39±0.33 | 95.79±0.20 | 84.56±2.36 |
| | EoA | 85.03±0.06 | 78.79±0.14 | 88.56±0.05 | 77.03±0.13 | 81.85±0.24 | 71.84±0.45 |
| | Mixup | 85.39±0.23 | 79.78±0.34 | 88.99±0.43 | 78.07±0.61 | 83.97±0.61 | 73.03±0.42 |
| | NCDG | 96.33±0.07 | 89.94±0.12 | 97.36±0.08 | 89.68±0.05 | 91.95±0.28 | 80.38±0.47 |
| | SAM | 98.27±0.15 | 88.84±0.32 | 99.37±0.06 | 91.65±0.41 | 97.29±0.19 | 73.03±2.53 |
| | UDIM | 98.39±0.09 | 89.67±0.95 | 99.19±0.10 | 92.15±0.41 | 96.69±0.25 | 71.10±1.19 |
| | D-BAT PL Ens | 98.67±0.02 | 89.48±0.08 | 99.23±0.05 | 96.39±0.07 | 99.01±0.24 | 66.69±0.13 |
| | XGB PL Ens | 98.73±0.14 | 65.40±0.52 | **99.00±0.04** | **92.67±0.33** | **99.57±0.01** | **96.93±0.09** |
| | EXPLOR D-BAT | 98.50±0.01 | 89.94±0.02 | 99.46±0.04 | 96.70±0.01 | 99.93±0.02 | 77.26±0.27 |
| | EXPLOR | **99.06±0.14** | 64.71±0.29 | 99.22±0.05 | 91.31±0.41 | 99.36±0.06 | 96.42±0.10 |
| AUPRC@ R<0.2 | DivDis* | 96.02±0.19 | 86.51±0.28 | 97.80±0.07 | 89.72±0.06 | 93.86±0.42 | 80.18±0.92 |
| | FixMatch* | 94.48±0.47 | 86.38±0.35 | 97.62±0.06 | 92.89±1.36 | 95.02±0.76 | 70.49±0.43 |
| | ERM | 96.15±0.16 | 88.67±0.23 | 99.04±0.04 | 91.44±0.43 | 96.95±0.11 | 72.88±0.95 |
| | D-BAT | 96.97±0.16 | 88.78±0.40 | 98.13±0.08 | 91.79±0.38 | 93.81±0.22 | **84.35±1.35** |
| | AdvStyle | 95.13±0.13 | 88.21±0.37 | 97.04±0.17 | 89.05±0.22 | 94.84±0.31 | 84.51±2.36 |
| | EoA | 85.03±0.06 | 78.79±0.14 | 88.56±0.05 | 77.03±0.13 | 81.85±0.24 | 71.84±0.45 |
| | Mixup | 85.39±0.23 | 79.78±0.34 | 88.99±0.43 | 78.07±0.61 | 83.97±0.61 | 73.04±0.42 |
| | NCDG | 93.92±0.62 | 84.46±0.67 | 97.83±0.08 | 87.82±0.26 | 93.40±0.78 | 80.09±1.94 |
| | SAM | 96.79±0.17 | 87.50±0.23 | 98.78±0.09 | 89.40±0.40 | 95.02±0.37 | 71.92±1.71 |
| | UDIM | 96.97±0.13 | 88.16±0.78 | 98.61±0.12 | 90.13±0.38 | 94.05±0.30 | 71.37±1.00 |
| | DBAT PL Ens | 96.67±0.06 | 88.11±0.04 | 98.74±0.07 | 94.05±0.08 | 97.51±0.21 | 69.93±0.04 |
| | XGB PL Ens | 98.00±0.07 | 89.48±0.24 | **99.14±0.02** | 94.20±0.15 | 97.79±0.11 | 68.48±0.30 |
| | EXPLOR DBAT | 97.78±0.04 | 88.97±0.02 | 99.10±0.16 | 94.52±0.02 | 98.25±0.01 | 75.50±0.15 |
| | EXPLOR XGB | **98.45±0.06** | **89.76±0.26** | 99.15±0.05 | **94.42±0.09** | **98.66±0.10** | 69.04±0.50 |
| AUPRC@ R<0.3 | DivDis* | 95.19±0.37 | 85.12±0.49 | 97.33±0.16 | 88.06±1.14 | 91.89±0.95 | 78.97±1.66 |
| | FixMatch* | 87.14±0.64 | 85.74±0.42 | 97.59±0.07 | 91.02±1.24 | 93.76±1.11 | 70.61±0.47 |
| | ERM | 95.06±0.11 | 87.42±0.13 | 98.69±0.05 | 89.86±0.39 | 95.58±0.15 | 71.98±0.84 |
| | D-BAT | 96.89±0.15 | 87.77±0.32 | 98.08±0.08 | 90.71±0.52 | 93.73±0.21 | **81.40±0.89** |
| | AdvStyle | 94.71±0.13 | 88.05±0.36 | 96.69±0.21 | 88.93±0.19 | 94.52±0.37 | 83.57±2.08 |
| | EoA | 85.03±0.06 | 78.79±0.14 | 88.56±0.05 | 77.03±0.13 | 81.85±0.24 | 71.84±0.45 |
| | Mixup | 85.39±0.23 | 79.78±0.34 | 88.99±0.43 | 78.07±0.61 | 83.97±0.61 | 73.06±0.43 |
| | NCDG | 95.17±0.04 | 86.07±0.08 | 95.89±0.05 | 86.53±0.03 | 88.73±0.19 | 76.05±0.35 |
| | SAM | 95.62±0.13 | 86.48±0.27 | 98.23±0.09 | 87.84±0.37 | 93.58±0.39 | 71.63±1.27 |
| | UDIM | 95.72±0.15 | 87.08±0.77 | 98.10±0.11 | 88.48±0.40 | 92.46±0.35 | 71.53±0.86 |
| | D-BAT PL Ens | 96.10±0.06 | 87.35±0.04 | 98.36±0.10 | 92.44±0.05 | 95.86±0.25 | 71.17±0.03 |
| | XGB PL Ens | 96.73±0.10 | 69.94±0.20 | 97.17±0.06 | 87.99±0.18 | 98.71±0.03 | 92.32±0.23 |
| | EXPLOR D-BAT | 97.01±0.01 | **87.87±0.02** | 98.61±0.07 | 92.78±0.02 | 96.27±0.01 | 74.76±0.10 |
| | EXPLOR XGB | **97.91±0.07** | 69.74±0.21 | **97.76±0.05** | 88.66±0.21 | **98.89±0.02** | **93.11±0.13** |
| AUPRC | DivDis* | 89.62±0.12 | 80.92±0.06 | 93.34±0.10 | 81.48±0.30 | 83.85±0.41 | *75.09±0.38* |
| | FixMatch* | 87.14±0.58 | 81.51±0.31 | 92.38±0.15 | 82.78±0.57 | 80.92±1.12 | 70.57±0.32 |
| | ERM | 89.84±0.07 | 82.26±0.07 | 94.72±0.07 | 82.59±0.21 | 87.70±0.09 | 70.95±0.44 |
| | D-BAT | 84.70±0.51 | 70.08±0.69 | 90.84±0.28 | 73.45±0.90 | 76.64±0.49 | 54.87±0.99 |
| | AdvStyle | 83.01±0.90 | 69.48±2.16 | 88.54±1.44 | 72.11±1.49 | 81.17±1.72 | 58.40±2.18 |
| | EoA | 69.66±0.47 | 57.71±0.79 | 79.12±0.09 | 56.52±0.42 | 64.16±0.51 | 36.50±1.48 |
| | Mixup | 80.36±0.88 | 72.88±1.67 | 86.88±0.14 | 74.99±0.15 | 73.03±1.67 | 60.84±4.31 |
| | NCDG | 89.27±0.22 | 80.54±0.21 | 94.17±0.08 | 81.19±0.11 | 87.72±0.37 | 74.99±0.74 |
| | SAM | 89.86±0.09 | 81.93±0.20 | 94.16±0.10 | 81.03±0.23 | 86.66±0.20 | 70.92±0.40 |
| | UDIM | 89.95±0.11 | 82.19±0.41 | 94.05±0.10 | 81.29±0.25 | 86.35±0.15 | 71.20±0.35 |
| | DBAT PL Ens | 91.19±0.08 | 82.72±0.04 | 94.79±0.10 | 84.11±0.01 | 88.36±0.28 | 72.13±0.07 |
| | XGB PL Ens | 91.21±0.02 | 82.61±0.05 | 94.91±0.03 | 84.13±0.06 | 88.44±0.05 | 71.80±0.07 |
| | EXPLOR D-BAT | 90.56±0.03 | 82.95±0.06 | 94.87±0.13 | 84.11±0.01 | 88.44±0.02 | **73.13±0.02** |
| | EXPLOR XGB | **91.59±0.03** | **83.06±0.10** | **95.38±0.01** | **84.77±0.02** | **89.52±0.05** | 71.41±0.11 |
| AUROC | DivDis* | 65.41±0.81 | 58.53±0.46 | 66.68±0.28 | 57.04±0.08 | 73.23±0.40 | 61.15±0.50 |
| | FixMatch* | 70.32±0.59 | 52.69±0.69 | 66.37±0.50 | 58.37±0.58 | 74.38±0.16 | 62.05±0.55 |
| | ERM | 67.73±0.18 | 59.15±0.23 | 77.41±0.24 | 62.68±0.31 | 72.24±0.09 | 52.32±0.59 |
| | D-BAT | 75.26±0.28 | 58.21±0.26 | 72.09±0.19 | 60.32±0.25 | **80.31±0.08** | 64.82±0.18 |
| | AdvStyle | 75.97±0.39 | 58.86±0.25 | 70.78±0.35 | 59.62±0.30 | 78.36±0.23 | 64.14±0.27 |
| | EoA | 64.91±0.34 | 52.71±0.44 | 59.27±0.20 | 54.63±0.24 | 62.99±0.16 | 55.83±0.18 |
| | Mixup | 68.20±0.66 | 56.33±0.45 | 60.39±0.40 | 56.50±0.37 | 64.24±1.23 | 57.75±0.80 |
| | NCDG | 73.66±0.53 | 57.70±0.81 | 67.20±0.33 | 57.18±0.18 | 76.48±0.22 | 61.44±0.13 |
| | SAM | 67.33±0.22 | 58.97±0.40 | 75.63±0.35 | 60.07±0.35 | 71.06±0.27 | 52.50±0.36 |
| | UDIM | 67.50±0.29 | 59.30±0.63 | 75.38±0.34 | 60.29±0.39 | 71.37±0.18 | 53.43±0.43 |
| | DBAT PL Ens | 69.40±0.06 | 60.04±0.31 | 77.83±0.40 | 64.66±0.04 | 74.51±0.48 | 56.39±0.20 |
| | XGB PL Ens | 73.70±0.07 | 56.48±0.06 | 70.17±0.04 | 59.77±0.02 | 77.78±0.09 | 64.89±0.06 |
| | EXPLOR DBAT | 70.64±0.04 | **59.90±0.03** | **78.52±0.37** | **64.67±0.01** | 75.34±0.07 | 56.32±0.03 |
| | EXPLOR XGB | **75.47±0.09** | 55.40±0.20 | 71.28±0.10 | 60.63±0.17 | 80.01±0.03 | **66.14±0.05** |

EXPLOR is assigning confident predictions to structurally diverse compounds rather than overfitting to a narrow subset of the chemical space.

Table 6: Full experiment results on ChEMBL (Gaulton et al., 2011) and Therapeutics Data Commons (Huang et al., 2021) datasets. We **bold** best scores based on the mean minus 1 standard deviation. Note that * refers to a semi-supervised method.

| | | hERG | A549_cells | cyp_2D6 | Ames |
|---|---|---|---|---|---|
| AUPRC R<0.1 | DivDis* | 86.92±8.44 | 89.01±3.53 | 83.65±6.17 | 97.47±1.68 |
| | FixMatch* | 84.77±1.81 | 92.93±3.00 | 92.71±1.98 | 96.26±1.04 |
| | ERM | 91.95±0.85 | 97.37±0.81 | 92.89±2.93 | 99.02±0.15 |
| | D-BAT | 88.55±1.68 | 98.57±0.16 | 95.71±0.89 | 99.07±0.23 |
| | AdvStyle | 93.27±0.56 | 96.89±0.30 | 84.21±2.07 | 99.52±0.12 |
| | EoA | 63.80±0.42 | 61.31±0.31 | 61.77±0.41 | 78.74±0.43 |
| | Mixup | 82.80±1.56 | 95.04±0.25 | 87.39±3.09 | 91.02±1.05 |
| | NCDG | 72.96±1.31 | 78.79±2.35 | 61.77±1.15 | 89.68±0.38 |
| | SAM | 90.69±1.67 | 97.22±0.63 | 90.20±0.46 | 98.29±0.19 |
| | UDIM | 91.03±2.36 | 97.39±0.42 | 88.30±1.14 | 97.99±0.44 |
| | D-BAT PL Ens | 86.29±1.00 | 98.38±0.23 | 96.66±0.63 | 1.00±0.00 |
| | XGB PL Ens | 96.65±0.22 | 99.74±0.05 | **99.81±0.11** | 98.73±0.27 |
| | EXPLOR D-BAT | 90.01±0.86 | 98.92±0.12 | 98.51±0.63 | 1.00±0.00 |
| | EXPLOR XGB | **98.10±0.34** | 99.76±0.03 | 99.48±0.17 | **99.66±0.20** |
| AUPRC@ R<0.2 | DivDis* | 85.65±2.37 | 89.45±1.16 | 79.98±1.10 | 95.80±0.34 |
| | FixMatch* | 79.66±0.57 | 90.81±4.30 | 83.58±2.55 | 93.55±0.64 |
| | ERM | 85.58±1.10 | 96.65±0.59 | 87.93±2.41 | 98.04±0.30 |
| | D-BAT | 84.48±1.74 | 98.26±0.14 | 91.40±0.98 | 99.04±0.24 |
| | AdvStyle | 88.21±0.77 | 97.77±0.28 | 84.83±0.93 | **99.05±0.17** |
| | EoA | 63.80±0.42 | 61.31±0.31 | 61.77±0.41 | 78.74±0.43 |
| | Mixup | 82.25±1.51 | 95.04±0.25 | 87.09±2.32 | 91.02±1.05 |
| | NCDG | 70.25±1.03 | 78.22±2.13 | 61.48±0.98 | 86.29±0.45 |
| | SAM | 84.37±1.06 | 95.70±0.75 | 85.02±0.69 | 96.66±0.38 |
| | UDIM | 85.07±2.32 | 95.87±0.36 | 84.80±0.84 | 96.53±0.61 |
| | DBAT PL Ens | 84.86±0.82 | 97.45±0.32 | 92.56±0.81 | 99.47±0.14 |
| | XGB PL Ens | 94.44±0.17 | 98.22±0.08 | 95.51±0.19 | 97.84±0.20 |
| | EXPLOR DBAT | 87.05±0.75 | 99.05±0.12 | 95.33±0.51 | **99.87±0.16** |
| | EXPLOR XGB | **94.67±0.29** | 98.87±0.09 | 96.88±0.25 | 98.45±0.19 |
| AUPRC@ R<0.3 | DivDis* | 83.69±3.39 | 88.83±2.50 | 77.55±1.02 | 94.87±0.74 |
| | FixMatch* | 76.16±1.04 | 89.12±5.07 | 77.70.58±2.05 | 92.72±0.51 |
| | ERM | 82.29±1.10 | 95.92±0.58 | 83.76±2.02 | 97.23±0.36 |
| | D-BAT | 82.44±1.59 | 97.37±0.24 | 87.65±0.58 | 98.61±0.24 |
| | AdvStyle | 85.05±0.86 | 96.47±0.29 | 82.76±0.95 | **98.71±0.20** |
| | EoA | 63.80±0.42 | 61.31±0.31 | 61.77±0.41 | 78.74±0.43 |
| | Mixup | 81.51±1.35 | 94.95±0.25 | 84.53±2.89 | 90.88±1.00 |
| | NCDG | 69.59±0.97 | 77.88±1.93 | 60.93±0.86 | 85.41±0.33 |
| | SAM | 80.55±0.72 | 94.18±0.61 | 82.66±0.68 | 95.24±0.36 |
| | UDIM | 80.87±1.85 | 94.67±0.46 | 82.03±0.38 | 95.36±0.73 |
| | D-BAT PL Ens | 84.13±0.59 | 96.97±0.14 | 89.06±1.24 | 98.77±0.10 |
| | XGB PL Ens | 90.46±0.06 | 97.35±0.05 | 92.34±0.24 | 97.83±0.15 |
| | EXPLOR D-BAT | 85.66±0.72 | 98.70±0.02 | 91.40±0.44 | 99.24±0.06 |
| | EXPLOR XGB | **90.88±0.34** | **97.96±0.1** | 93.06±0.27 | 98.18±0.15 |
| AUPRC | DivDis* | 67.70±0.25 | 76.45±0.63 | 65.76±0.45 | 82.37±0.27 |
| | FixMatch* | 45.16±3.11 | 51.31±1.28 | 32.65±2.16 | 72.54±1.15 |
| | ERM | 68.77±0.40 | 81.73±0.37 | 67.97±0.81 | 86.01±0.23 |
| | D-BAT | 54.60±1.59 | 67.04±0.53 | 47.42±0.85 | 70.44±0.76 |
| | AdvStyle | 51.54±0.93 | 65.02±0.72 | 44.41±0.96 | 74.98±0.49 |
| | EoA | 43.30±0.51 | 44.95±0.24 | 37.37±0.95 | 59.43±0.18 |
| | Mixup | 42.42±0.85 | 50.52±0.50 | 27.79±1.49 | 60.94±0.80 |
| | NCDG | 42.69±0.24 | 49.13±0.79 | 31.08±0.16 | 67.79±0.93 |
| | SAM | 66.89±0.28 | 79.88±0.34 | 67.37±0.31 | 82.50±0.39 |
| | UDIM | 67.23±0.60 | 79.98±0.37 | 66.86±0.43 | 82.97±1.05 |
| | DBAT PL Ens | 72.22±0.18 | 83.79±0.31 | 73.26±0.65 | 89.08±0.11 |
| | XGB PL Ens | 72.19±0.07 | 84.10±0.01 | 72.93±0.13 | 87.43±0.02 |
| | EXPLOR DBAT | 72.73±0.33 | 84.80±0.11 | 73.64±0.45 | **89.38±0.06** |
| | EXPLOR XGB | **73.26±0.08** | **84.60±0.08** | **73.59±0.15** | 88.50±0.10 |
| AUROC | DivDis* | 71.19±0.57 | 72.20±0.41 | 64.50±0.77 | 77.05±0.47 |
| | FixMatch* | 68.41±0.84 | 61.16±1.31 | *80.85±0.40* | 70.32±0.59 |
| | ERM | 73.58±0.23 | 76.58±0.30 | 65.25±0.55 | 82.02±0.26 |
| | D-BAT | 76.58±0.45 | 78.16±0.23 | 67.54±0.47 | 83.82±0.15 |
| | AdvStyle | 75.84±0.46 | 76.13±0.28 | 65.51±0.62 | **85.56±0.71** |
| | EoA | 68.02±0.34 | 68.33±0.24 | 60.50±0.48 | 74.77±0.25 |
| | Mixup | 73.96±0.26 | 76.57±0.42 | 67.53±0.90 | 78.43±0.49 |
| | NCDG | 70.42±0.13 | 72.50±0.79 | 62.01±0.16 | 73.42±0.93 |
| | SAM | 72.17±0.41 | 74.67±0.49 | 64.45±0.53 | 78.51±0.47 |
| | UDIM | 73.17±0.38 | 74.64±0.38 | 64.09±0.66 | 78.63±1.30 |
| | DBAT PL Ens | 76.51±0.07 | 78.34±0.35 | **70.80±0.36** | 84.37±0.08 |
| | XGB PL Ens | 74.74±0.06 | 79.17±0.02 | 70.33±0.07 | 81.87±0.04 |
| | EXPLOR DBAT | **76.84±0.10** | 79.27±0.14 | 70.11±0.33 | **85.18±0.04** |
| | EXPLOR XGB | 75.97±0.08 | **79.53±0.06** | 70.54±0.42 | 83.78±0.12 |

Table 7: Full experiment results on Tableshift (Gardner et al., 2023) datasets. We **bold** best scores based on the mean minus 1 standard error. Note that * refers to a semi-supervised method that uses additional unlabeled OOD data during training.

| | | Childhood Lead | FICO HELOC | Hospital Readmission | Sepsis |
|---|---|---|---|---|---|
| AUPRC@ R<0.1 | DivDis* | 91.67±3.03 | 84.88±3.80 | 88.99±3.36 | 59.50±1.55 |
| | FixMatch* | 86.68±3.32 | 83.99±2.27 | 73.97±2.64 | 17.84±1.87 |
| | ERM | 54.24±0.00 | 85.84±0.88 | **90.52±0.36** | 17.35±0.94 |
| | D-BAT | 52.67±0.00 | 92.97±0.59 | 83.73±0.12 | 81.99±0.33 |
| | AdvStyle | 63.63±0.00 | 90.10±1.03 | 77.29±0.74 | 60.79±0.72 |
| | EoA | 75.73±0.83 | 63.42±2.13 | 52.03±1.78 | 43.76±1.14 |
| | Mixup | 50.00±0.00 | 92.79±0.03 | 85.71±0.35 | 68.21±1.28 |
| | NCDG | 83.30±0.20 | 91.70±0.33 | 70.42±0.17 | 72.40±0.04 |
| | SAM | 97.50±0.00 | 94.42±1.07 | 80.13±0.57 | 66.42±1.20 |
| | UDIM | 97.06±0.45 | 95.07±0.92 | 79.59±1.32 | 65.62±0.53 |
| | D-BAT PL Ens | 98.72±0.42 | 96±0.05 | 76.47±0.05 | 75.32±0.01 |
| | XGB PL Ens | 98.69±0.03 | 92.18±0.58 | 51.31±0.16 | **82.85±1.53** |
| | EXPLOR D-BAT | 98.36±0.05 | 96.99±0.06 | 81.84±0.34 | 77.25±0.17 |
| | EXPLOR XGB | **99.72±0.10** | **93.93±0.87** | 66.65±0.17 | 80.74±1.77 |
| AUPRC@ R<0.2 | DivDis* | 89.77±2.51 | 85.50±2.65 | 83.22±2.57 | 60.66±1.57 |
| | FixMatch* | 81.75±3.69 | 77.76±1.94 | 67.84±1.95 | 17.28±0.58 |
| | ERM | 43.66±0.00 | 85.74±0.78 | **84.35±0.28** | 15.31±0.53 |
| | D-BAT | 62.82±0.00 | 91.20±0.24 | 78.84±0.12 | 75.37±0.38 |
| | AdvStyle | 64.96±0.01 | 88.71±0.65 | 72.91±0.58 | 59.83±0.63 |
| | EoA | 77.43±0.97 | 59.53±2.24 | 51.83±1.66 | 41.10±1.56 |
| | Mixup | 50.00±0.00 | 91.16±2.10 | 69.19±3.95 | 66.09±1.15 |
| | NCDG | 82.49±0.18 | 90.93±0.31 | 68.88±0.15 | 70.22±0.03 |
| | SAM | 95.00±0.00 | 91.42±0.69 | 74.13±0.49 | 65.34±0.73 |
| | UDIM | 94.11±0.90 | 92.58±0.67 | 73.65±0.67 | 65.09±0.28 |
| | DBAT PL Ens | 93.81±0.03 | 92.87±0.05 | 73.52±0.37 | 72.74±0.13 |
| | XGB PL Ens | 97.39±0.06 | 90.07±0.32 | 58.30±0.09 | **78.62±1.57** |
| | **EXPLOR** (D-BAT) | 94.69±0.02 | **93.33±0.02** | 77.20±0.23 | 73.89±0.08 |
| | **EXPLOR** (XGB) | **97.92±0.20** | 91.72±0.62 | 67.57±0.12 | 76.95±1.45 |
| AUPRC@ R<0.3 | DivDis* | 87.90±2.20 | 85.77±2.73 | 79.82±2.42 | 61.55±1.55 |
| | FixMatch* | 81.61±3.09 | 76.48±4.13 | 64.47±1.70 | 18.45±0.46 |
| | ERM | 38.83±0.00 | 85.25±0.60 | **80.54±0.25** | 14.41±0.35 |
| | D-BAT | 68.11±0.00 | 90.11±0.13 | 75.26±0.10 | 70.92±0.44 |
| | AdvStyle | 68.67±0.08 | 87.33±1.21 | 70.27±0.49 | 59.02±0.57 |
| | EoA | 79.88±1.03 | 56.52±2.34 | 51.34±1.27 | 40.09±1.05 |
| | Mixup | 50.00±0.00 | 90.01±0.02 | 47.18±3.54 | 64.41±0.95 |
| | NCDG | 82.10±0.15 | 88.76±0.29 | 67.52±0.13 | 69.48±0.03 |
| | SAM | 92.50±0.00 | 89.90±0.61 | 71.03±0.44 | 64.28±0.57 |
| | UDIM | 91.17±1.36 | 91.06±0.44 | 70.66±0.60 | 64.24±0.23 |
| | D-BAT PL Ens | 91.71±0.04 | 91.52±0.01 | 71.78±0.27 | 70.33±0.16 |
| | XGB PL Ens | 96.08±0.08 | 89.73±0.23 | 60.86±0.04 | **75.84±1.55** |
| | EXPLOR D-BAT | 92.60±0.03 | 91.72±0.07 | 72.69±0.18 | 70.92±0.06 |
| | EXPLOR XGB | **96.58±0.22** | **91.10±0.41** | 67.36±0.06 | 74.02±1.20 |
| AUPRC | DivDis* | 76.33±0.86 | 82.47±1.05 | 65.95±2.15 | 58.85±1.21 |
| | FixMatch* | 75.39±0.58 | 72.22±4.46 | 56.54±1.21 | 18.24±0.66 |
| | ERM | 23.60±0.11 | 77.37±0.27 | **67.56±0.13** | 11.12±0.12 |
| | D-BAT | 71.85±0.01 | 80.91±0.17 | 63.29±0.08 | 58.17±0.21 |
| | AdvStyle | 48.37±2.77 | 79.63±1.65 | 38.13±3.80 | 54.34±0.27 |
| | EoA | 49.48±0.19 | 59.53±2.24 | 29.45±4.95 | 11.21±2.21 |
| | Mixup | 50.00±0.00 | 80.95±0.63 | 14.15±1.06 | 56.80±0.40 |
| | NCDG | 73.08±0.14 | 79.05±0.21 | 58.95±0.10 | 61.96±0.02 |
| | SAM | 75.00±0.00 | 81.61±0.43 | 61.12±0.28 | 57.90±0.22 |
| | UDIM | 73.70±1.38 | 82.10±0.44 | 60.95±0.33 | 57.97±0.17 |
| | DBAT PL Ens | 82.72±0.01 | 81.35±0.03 | 62.34±0.14 | 59.97±0.10 |
| | XGB PL Ens | 86.39±0.19 | 83.80±0.07 | 62.83±0.03 | **64.01±0.94** |
| | **EXPLOR** (D-BAT) | 82.76±0.02 | 81.37±0.04 | 63.30±0.14 | 50.97±0.05 |
| | **EXPLOR** (XGB) | **86.70±0.28** | **84.02±0.09** | 63.62±0.08 | 62.21±0.52 |
| AUROC | DivDis* | 77.74±1.90 | 83.55±1.06 | 65.28±0.59 | 62.33±0.61 |
| | FixMatch* | 79.87±0.29 | 74.29±1.69 | 55.78±0.98 | 49.50±1.01 |
| | ERM | 78.41±0.14 | 73.71±0.17 | **67.06±0.10** | 62.79±0.14 |
| | D-BAT | 79.13±0.02 | 76.13±0.03 | 63.22±0.03 | 57.95±0.04 |
| | AdvStyle | 74.45±0.04 | 77.23±1.69 | 61.32±0.34 | 55.31±0.34 |
| | EoA | 72.62±0.32 | 54.67±2.55 | 51.65±1.70 | 49.14±2.60 |
| | Mixup | 50.00±0.00 | 78.74±0.17 | 63.37±0.25 | 56.82±0.26 |
| | NCDG | 76.91±0.13 | 75.09±0.79 | 58.67±0.16 | **63.41±0.93** |
| | SAM | 50.00±0.00 | 79.70±1.64 | 61.14±0.32 | 58.71±0.23 |
| | UDIM | 54.50±4.18 | 79.41±1.07 | 61.05±0.32 | 58.87±0.24 |
| | DBAT PL Ens | 84.89±0.05 | 76.32±0.03 | 63.16±0.10 | 59.23±0.04 |
| | XGB PL Ens | 84.88±0.19 | **83.53±0.03** | 63.18±0.03 | 62.53±0.82 |
| | **EXPLOR** (D-BAT) | 84.36±0.23 | 76.27±0.01 | 63.51±0.16 | 59.01±0.07 |
| | **EXPLOR** (XGB) | **87.95±0.25** | 83.11±0.04 | 63.65±0.07 | 61.42±0.35 |

Table 8: ChEMBL Datasets' Mean AUPRC Ablating Type of pseudo-labelers.

| AUPRC@R<0.2 | XGBoost | Random Forest | Decision Tree |
|---|---|---|---|
| PL Ens | 95.87 | 95.72 | 92.18 |
| EXPLOR | 96.64 | 96.15 | 94.15 |

### C.4 FULL EXPERIMENT RESULTS ON TABLESHIFT

In Table 7, we report the full results on full results on Tableshift (Gardner et al., 2023) datasets.

### C.5 PSEUDO-LABELERS ABLATIONS

In this section, We ablate the kind of pseudo-labelers as EXPLOR relies on pseudo-labels during training. reports the mean AUPRC@R<0.2 across ChEMBL datasets when using XGBoost, Random Forest (with 100 estimators), and Decision Tree as pseudo-labelers. Performance was comparable between XGBoost and Random Forest, indicating robustness to the choice of strong ensemble models. In contrast, using a weaker pseudo-labelers such as a single Decision Tree led to a performance drop. Nevertheless, EXPLOR consistently outperformed its respective pseudo-labelers, demonstrating its ability to enhance predictions regardless of pseudo-labeler strength (Tab. 8.

### C.6 PER-HEAD MATCHING ABLATION DETAILS

In this section we provide details on the per-head matching ablations where we ablate the matching loss scheme on pseudo-labelers and explore a mean-only matching approach on expanded points as an alternative. First, we consider utilizing a single-headed (SH) MLP, $f(x)$ ($512 \rightarrow 512 \rightarrow 1$), which is trained via a mean matching loss $\mathcal{L}_{\mathrm{MM}}(f, \{g_j\}_{j=1}^K; \mathcal{S}) \equiv \frac{1}{|\mathcal{S}|} \sum_{x \in S} \ell(f(x), \frac{1}{K} \sum_{j=1}^K g_j(x))$, rather than the per-expert matching loss, $\mathcal{L}_{\mathrm{match}}$ equation 3 We also explored the effect of training our multi-headed (MH) architecture ($512 \rightarrow 512 \rightarrow 1024$) using only mean-matching (without per-head matching), $\mathcal{L}'_{\mathrm{MM}}(\{h_j\}_{j=1}^K, \{g_j\}_{j=1}^K; \mathcal{S}) \equiv \frac{1}{|\mathcal{S}|} \sum_{x \in S} \ell(\frac{1}{K} \sum_{j=1}^K \sigma(h_j(x)), \frac{1}{K} \sum_{j=1}^K g_j(x))$.

### C.7 BOTTLENECK ABLATIONS DETAILS

Table 9: ChEMBL datasets' mean AUPRC ablating hidden layer and output sizes.

| | @R<.2 | @R<1 |
|---|---|---|
| Full | 97.12 | 77.28 |
| Full (ML) | 96.94 | 77.25 |
| Full (ERM) | 92.65 | 72.61 |
| Tiny | 96.29 | 76.63 |
| Tiny (ML) | 96.33 | 76.17 |
| Tiny (ERM) | 88.28 | 70.87 |

In this section, we provide the details and results for the bottleneck ablations. Results are shown in Tab. 9, here 'Full' denotes our original $2 \times 512$ hidden layer architecture, where 'tiny' denotes a $2 \times 32$ hidden layer architecture (a $\times 16$ decrease in parameters). Moreover, we also consider mean logits 'ML,' a final averaging over the multi-head logits that produces a single output unit (i.e., averaging the output weights/bias after training to construct the mean logits network). The performance gap is marginal between the 'Full' and 'Tiny' model (a $0.85\%$ difference) when using our proposed loss. In contrast, when using empirical risk minimization, we see a $4.87$ times bigger drop in performance between 'Full' and 'Tiny' models. This suggests that the bottlenecking properties of our method are key to EXPLOR's performance. Moreover, the results show promise for EXPLOR in resource-constrained settings (such as in IoT applications).

## D ADDITIONAL FIGURES

Predicted probabilities from EXPLOR network and pseudo-labelers. We highlight example instances where the pseudo-labelers initially makes incorrect predictions but are corrected when we average the predicted probabilities from EXPLOR network and EXPLOR base.

# E   LIMITATIONS

While EXPLOR consistently outperforms its pseudo-labelers across diverse model types (e.g., XG-Boost, Random Forest, Decision Tree, D-BAT), its effectiveness can be influenced by the quality of the pseudo-labeler and the learned latent space. However, our results show that EXPLOR remains robust even with simpler pseudo-label models and standard latent representations, suggesting room for further gains with more sophisticated choices. To maintain broad applicability, our experiments were constrained to real-valued vector data with general augmentations. In domain-specific applications, incorporating modality-aware augmentations could further enhance performance. Future work may explore this direction to extend EXPLOR's effectiveness.

