# OpenReview forum: "EXPLOR: Extrapolatory Pseudo-Label Matching for OOD Uncertainty Based Rejection"
_ICLR.cc/2026/Conference — Submitted to ICLR 2026_

### Official Review · Reviewer_mPjQ · 2025-10-30

**Soundness:** 4
**Presentation:** 4
**Contribution:** 2
**Rating:** 4
**Confidence:** 4

**Summary:**

This paper introduces EXPLOR, a framework for single-source out-of-distribution (OOD) generalization and uncertainty-based rejection. The method integrates three elements: (1) latent-space extrapolation to synthetically expand training support, (2) supervision from a diverse set of pseudo-labelers, and (3) a multi-head student trained with per-head and mean-matching losses. A bias–variance decomposition is provided as an intuitive justification. Experiments on seven chemical and several tabular datasets demonstrate consistent improvements in AUPRC and AUPRC@R<τ over relevant baselines.

**Strengths:**

- **Relevance and clarity**
The paper addresses an important problem in OOD generalization under a single-source assumption. The motivation is clear, the method is concise, and the presentation is easy to follow.

- **Methodological consistency**
The combination of per-head matching and latent extrapolation provides a coherent multi-task formulation that plausibly enhances robustness.

- **Reproducibility and efficiency**
The implementation is simple, cost-efficient, and reproducible with standard hardware.

**Weaknesses:**

- **Redundancy at inference**
EXPLOR retains all pseudo-labelers at test time (Eq. 8), forming a hybrid ensemble of teachers and student heads. This design questions whether the student truly distills ensemble knowledge or merely supplements it. The claimed variance-reduction interpretation (Eq. 9–10) remains heuristic without quantitative analysis.

- **Limited conceptual novelty**
Each component of EXPLOR—ensemble pseudo-labeling, latent-space augmentation, and mean-matching regularization—has clear precedents in prior work (e.g., Hydra 2021, PixMix 2022, ACET 2019). The paper’s main contribution lies in system-level integration within a single-source OOD setup rather than in a new theoretical insight.

- **Dataset and representation scope**
The use of ChEMBL, TDC, and DrugOOD benchmarks is reasonable, but all experiments employ fixed ECFP4 fingerprints. This limits the generality of the “modality-agnostic” claim. Results on learned embeddings, such as graph neural networks or visual features, would better demonstrate transferability.

- **Ablation and statistical support**
Per-head matching and bottleneck ablations show small absolute gains (around 1 percent) and lack significance testing. The uncertainty claims would be stronger with calibration or variance metrics (for example expected calibration error).

- **Heuristic theoretical argument**
The bias–variance decomposition provides intuition but lacks empirical verification. No analysis is presented to confirm that predictive variance actually decreases as claimed.

**Questions:**

**Inference overhead**
What is the actual computational overhead in terms of time and memory when retaining up to 1024 pseudo-labelers during inference compared with using the student alone? A quantitative comparison would clarify the method’s practicality.

**Dependence on pseudo-labelers**
How would EXPLOR perform if pseudo-labelers were unavailable at test time? Would the student alone preserve similar accuracy and uncertainty behavior?

**Generality across learned embeddings**
Does the proposed framework extend beyond fixed handcrafted features to learned embeddings such as those obtained from graph neural networks or visual backbones? Showing this would strengthen the claim of modality-agnostic generalization.

**Notation clarification for Equation 8**
Equation 8 appears to omit parentheses in the summation term.

---

> ### Author Response · Authors · 2025-11-21
>
> Thank you for your time and thoughtful comments. We are encouraged to see that you appreciated our relevance and clarity, coherent multi-task formulation, and efficiency. Please see our response below.
>
> **Novelty**   Please see general response on novelty and contribution. Foundational advances such as Variational Autoencoders (variational inference + autoencoders), U-Net (CNNs + skip connections), and Transformers (attention + feed-forward networks + residual connections) were considered novel because their combinations produced behaviors that none of the components could achieve alone. In the same spirit, EXPLOR’s *novel combination* of ensemble learning, (*a novel expansive*) latent-space augmentation, and mean-matching regularization together with *the novel multihead matching training* yields a meaningfully new training framework tailored to the single-source domain generalization setting—a scenario where existing techniques do not directly apply.
>
> **Redundancy and Inference**  Although the pseudo-labeler ensemble (PL) is slower than a single NN, the absolute cost remains modest: on ChEMBL (average sample size of 1000 test samples), full PL inference completes in 2.37 seconds (w/o any code optimization), which is practical for scientific workflows we target. Importantly, if evaluation time is a priority, the EXPLOR multi-head neural network can be used on its own, requiring only 0.0124 seconds under the same setup. The neural network on its own also performs competitively—for example, 93.91 vs. 94.67 on hERG, 98.64 vs. 98.87 on A549—while EXPLOR offers additional robustness and stronger OOD behavior when the PL ensemble is used at inference.
>
> **Variance Reduction Justification**     Please see our general comments above on justification of our motivation and results on Table 3 and Figure 6, which validate the variance reduction properties of EXPLOR. We see that our EXPLOR loss substantially reduces the prediction variance on OOD points across bootstrap-type trials as compared to empirical risk minimization.
>
> **Representation Scope and Features**      Please note that our Tableshift experiments (Table 2) were performed on a *diverse set of non molecular fingerprint features* from various nonchemical domains.
>
> **Eq 8**     We are happy to add parentheses for additional clarity.

---

### Official Review · Reviewer_8PYw · 2025-11-01

**Soundness:** 3
**Presentation:** 3
**Contribution:** 2
**Rating:** 4
**Confidence:** 3

**Summary:**

This paper proposes a new method for single-source domain generalization. The method trains multiple pseudo-labelers on different data subsets. It expands training data through latent-space augmentations. Then it trains a multi-headed network where each head matches a different pseudo-labeler. The approach works with any vector data and different base models. Experiments show strong performance on high-confidence predictions.

**Strengths:**

1. Single-source domain generalization is realistic for drug discovery applications. Many real scenarios only have one labeled dataset. The paper targets this practical setting.

2. Works with tree models and neural networks. Works with any vector data. Unlike image-specific methods, this is general purpose. Can be applied to many domains.

3. Tests on diverse datasets including chemical and tabular data. Compares with multiple baselines. Shows consistent improvements over pseudo-labelers across datasets.

**Weaknesses:**

1.  The expansion pushes points away from origin by z' = (1 + |ε|)z. But no validation that expanded points are realistic. They might just be random noise. No check if they represent plausible OOD samples.

2. The method relies heavily on pseudo-labels from diverse labelers. But no check if pseudo-labels are reliable on OOD data. If all pseudo-labelers are wrong, student learns bad supervision. No quality control mechanism.

3. Diverse ensembles are standard ensemble practice. Pseudo-labeling is well-known in semi-supervised learning. Main contribution is combining them with latent expansion. The individual components are not new.

**Questions:**

1. How do you verify expanded points are realistic? Can you show they resemble real OOD samples?

2. How do you ensure pseudo-labels are reliable? What if all pseudo-labelers are wrong?

3. Why 1024 models? How did you choose this number? What happens with 64 or 256?

---

> ### Author Response · Authors · 2025-11-21
>
> Thank you for your time and thoughtful comments. We are encouraged to see that you appreciated our realistic setting, ability to work with general vector data, and our diverse experimentations. Please find our response below.
>
> **Noise Model**    Please note that the problem of generating “realistic samples” in a *general* single-source setting is actually ill-posed (as it is not possible to predict what is plausible outside of the data support for a blackbox ID sample). Thus, our concern is not to general “realistic samples” but instead to provide useful additional signals in areas outside of the typical support, which we demonstrate with the additional performance gain through our training. Naïvely perturbing inputs in raw feature space could generate unrealistic samples. However, EXPLOR does not operate in raw input space; all expansions are performed in a standardized embedding space produced by the shared backbone. The transformation z′=(1+∣ϵ∣)z therefore does not inject arbitrary noise but performs a direction-preserving radial expansion, increasing magnitude while preserving the semantic direction of the original embedding. The purpose of the expansion is therefore not to generate literal target-domain instances, but to explore nearby, higher-uncertainty regions of the latent space where the model is less confident. This produces informative synthetic points that improve robustness, as confirmed by our ablations.
>
> **Pseudo-labeler Performance**    Please note that the performance of pseudo-lablers on OOD was directly reported in our experimental results as “D-BAT PL Ens” and “XGB PL Ens” (Tables 1 and 2), showing that they are reliable on OOD data. Please note also (line 283) that we consider 64 DBat Pseudo-lablers.
>
> **Novelty**   Please see general response on novelty and contribution. Foundational advances such as Variational Autoencoders (variational inference + autoencoders), U-Net (CNNs + skip connections), and Transformers (attention + feed-forward networks + residual connections) were considered novel because their combinations produced behaviors that none of the components could achieve alone. In the same spirit, EXPLOR’s *novel combination* of ensemble learning, (*a novel expansive*) latent-space augmentation, and mean-matching regularization together with *the novel multihead matching* training yields a meaningfully new training framework tailored to the single-source domain generalization setting—a scenario where existing techniques do not directly apply.

---

### Official Review · Reviewer_EE2A · 2025-11-01

**Soundness:** 3
**Presentation:** 3
**Contribution:** 3
**Rating:** 6
**Confidence:** 3

**Summary:**

This paper introduces EXPLOR, a method for single-source out-of-distribution (OOD) generalization using pseudo-label matching across multiple heads.
The core idea is to expand latent representations beyond the in-distribution manifold via simple scaling, generate pseudo-labels from diverse models (e.g., XGBoost, D-BAT), and train a multi-head network to match each pseudo-labeler’s predictions.
The approach aims to improve high-confidence OOD predictions and rejection accuracy.
Experiments on chemical (ChEMBL, DrugOOD) and tabular (Tableshift) datasets show consistent improvements over baseline and semi-supervised methods.

**Strengths:**

1. Clear problem formulation. The paper targets the single-source OOD setting, which is realistic but rarely addressed. The motivation from drug screening and risk prediction is well justified.

2. Simple yet effective design. EXPLOR combines latent-space expansion and per-head pseudo-label matching into a lightweight framework that requires no unlabeled OOD data or domain annotations.

3. Strong empirical performance. Across more than ten datasets, EXPLOR achieves stable gains, particularly in the high-confidence regime (AUPRC@R < 0.2).

4. New evaluation metric. The introduction of AUPRC@R < τ provides a practical way to assess selective prediction quality, relevant to safety-critical tasks.

**Weaknesses:**

1. Limited novelty. The method essentially combines self-training, ensemble averaging, and multi-task learning in a new context.
No fundamentally new theoretical idea or architecture is introduced.

2. Empirical generality overstated. Although the paper claims to be modality-agnostic, all experiments are confined to tabular data.
There is no evidence that the approach extends to images, text, or graphs.

3. Dependence on pseudo-label quality. The performance gain scales with the accuracy of pseudo-labelers. Poor labelers could degrade the overall results, and this dependency is only briefly mentioned in the appendix.

4. Weak theoretical justification. The variance-reduction derivation (Eq. 9–10) is heuristic; there is no formal analysis showing that latent scaling approximates the true OOD distribution.

5. Overstated terminology. Terms like “extrapolatory” or “modality-agnostic” may mislead readers, given the limited scope of experiments.

**Questions:**

Please carefully read Weakness and answer all the five concerns.

---

> ### Author Response · Authors · 2025-11-21
>
> Thank you for your time and thoughtful comments. We are encouraged to see that you appreciated our realistic setting, novel metrics, strong performance, and effective design. Please find our responses below.
>
> **Novelty**     Please see general response on novelty and contribution. Foundational advances such as Variational Autoencoders (variational inference + autoencoders), U-Net (CNNs + skip connections), and Transformers (attention + feed-forward networks + residual connections) were considered novel because their combinations produced behaviors that none of the components could achieve alone. In the same spirit, EXPLOR’s *novel combination* of ensemble learning, (*a novel expansive*) latent-space augmentation, and mean-matching regularization together with *the novel multihead matching* training yields a meaningfully new training framework tailored to the single-source domain generalization setting—a scenario where existing techniques do not directly apply.
>
>
> **Modality-agnostic**    We meant that the components of  EXPLOR do not rely on spatial or sequential structure, in contrast with image- or text-specific augmentation pipelines. This is precisely why the augmentation mechanism is modality-agnostic in design: it perturbs latent features in a way that applies equally to tabular vectors, or any fixed-length learned representation.
>
> **Pseudo-Labeler**    A gain scaling with pseudo-labler quality is a feature rather than a bug. Our work is largely orthogonal to much of the other OOD generalization approaches and may be combined with any base learner that employs strong learning techniques.
>
> **Justification**   Please see our general comments above on justification of our motivation and results on Table 3 and Figure 6, which validate the variance reduction properties of EXPLOR.
>
> **Terminology**   Please note that we explicitly define “extrapolation to encompass prediction outside of the training data distribution support” akin to notions of OOD in our footnote in page 1. We are happy to elaborate on these terms in a camera ready copy.

---

### Author Response · Authors · 2025-11-21

We sincerely thank all the reviewers for their insightful comments and suggestions on our manuscript. We are pleased that the reviewers found EXPLOR’s novel approach well motivated, and appreciated its strong empirical performance as a single source domain generalization method on a diverse, thorough evaluation.

**Contributions and Novelty**

EXPLOR leverages concepts such as pseudo-labeling, ensemble learning, and latent-space augmentation, however, its core mechanism fundamentally differs from prior work. In EXPLOR, the multi-head architecture shares a common feature extractor, but branches into multiple specialized prediction heads that are each trained on different views of the data. These heads therefore learn complementary hypotheses over a shared representation, creating latent structure that cannot be reproduced by simply stacking existing techniques and that enables OOD generalization without multiple environments, which is crucial for single source domain generalization setting where only one environment is available. While EXPLOR is inspired by prior components—as is common in ML research—its contribution lies in the capability enabled by their integration, not in any individual part. Foundational advances such as Variational Autoencoders (variational inference + autoencoders), U-Net (CNNs + skip connections), and Transformers (attention + feed-forward networks + residual connections) were considered novel because their combinations produced behaviors that none of the components could achieve alone. In the same spirit, EXPLOR’s *novel combination* of ensemble learning, (*a novel expansive*) latent-space augmentation, and mean-matching regularization together with *the novel multihead matching* training yields a meaningfully new training framework tailored to the single-source domain generalization setting—a scenario where existing techniques do not directly apply.


**Empirical Studies**

Our intention in using the term “modality-agnostic” is not to claim universal validation across all modalities, but rather to clarify that EXPLOR operates independently of modality-specific inductive biases (e.g., convolutions for images, token positions for text). In our paper, we used  *ECFP4 fingerprints* for TDC, ChEMBL, and DrugOOD experiments, and *tabular data* from tableshift to evaluate the model as 2 different modality to validate our claim. **Note that the tabular data experiments operate over a wide range of general features in various domains**. The components of  EXPLOR – multi-head model supervised by model-agnostic pseudo-labelers and latent-space augmentation do not rely on spatial or sequential structure, in contrast with image- or text-specific augmentation pipelines. This is precisely why the augmentation mechanism is modality-agnostic in design: it perturbs latent features in a way that applies equally to tabular vectors, or any fixed-length learned representation. Moreover, EXPLOR can be paired with any pseudo-labelers, including tree-based predictors, MLPs, and neural encoders such as GNNs or vision transformers, so it is modality-agnostic by design. See Table 7 in Appx. for an ablation study that shows that EXPLOR improves the performance over myriad pseudo-lablers.


**Theoretical Motivation Justification**

One motivation of our methodological design is through the lens of variance reduction (eq. 9). We conducted additional experiments to confirm the variance reduction properties of EXPLOR as detailed in Fig. 6 and Table 3. By performing bootstrap-type trials we compared the variance of predictions on OOD points across trials for EXPLOR models versus the empirical risk minimization (ERM) loss. We see a significant reduction of variance, further validating our design and motivation.

**Summary**

In summary, we believe that EXPLOR’s approach for handling single source domain generalization problems will be valuable to the machine learning community, and thus hope it is recommended for publication.

---

### Meta-Review · Area_Chair_8ydy · 2026-01-21

**Summary:**

The submission proposes a novel framework for building models that can generate high-precision predictions for OOD examples. Reviewers consistently commend the clarity of the formulation and the practical experimental setup. Two recurring concerns, however, informed the decision. First, reviewers note that the core techniques are built on existing approaches, raising questions about novelty. Second, reviewers also express concern about the claim on modality-agnostic nature of the framework: the approach relies on feature-space perturbations to construct OOD examples yet the role and suitability of the feature extractor are not sufficiently explored for more complex modalities such as text and vision. While the rebuttal adequately addresses the novelty concern,  the evidence remains insufficient to support the generalization of the approach across modalities. Based on the unresolved issue, the AC recommends rejection.

**Reviewer Concerns:**

Weaknesses:
[W1] Novelty - All reviewers express concerns that the framework is largely based on existing approaches (e.g., self-training and multi-tasking).

[W2] Claim on modality agnostic - All reviewers expressed concerns on whether the proposed framework is modality agnostic, specifically, the validity of OOD examples and the lack of experiments for more modality

[W3] Weak theoretical justifications - two reviewers (mPjQ, EE2A) express concerns regarding the theoretical grounding of the approach.

The AC finds that W1 and W3 are sufficiently addressed in the rebuttal; the evidence for W2 remains lacking.

**Reviewer Scores:**

Due to W2, the AC thinks it would be difficult for reviewers to raise their initial ratings had they been able to participate fully in the discussion.

---

### Decision · Program_Chairs · 2026-01-26

Reject